## REVIEW ARTICLE

# Immunological considerations and challenges for regenerative cellular therapies

Sandra Petrus-Reurer [1,4✉], Marco Romano [2,4], Sarah Howlett[3], Joanne Louise Jones [3], Giovanna Lombardi [2] & Kourosh Saeb-Parsy [1✉]

The central goal of regenerative medicine is to replace damaged or diseased tissue with cells that integrate and function optimally. The capacity of pluripotent stem cells to produce unlimited numbers of differentiated cells is of considerable therapeutic interest, with several clinical trials underway. However, the host immune response represents an important barrier to clinical translation. Here we describe the role of the host innate and adaptive immune responses as triggers of allogeneic graft rejection. We discuss how the immune response is determined by the cellular therapy. Additionally, we describe the range of available in vitro and in vivo experimental approaches to examine the immunogenicity of cellular therapies, and finally we review potential strategies to ameliorate immune rejection. In conclusion, we advocate establishment of platforms that bring together the multidisciplinary expertise and infrastructure necessary to comprehensively investigate the immunogenicity of cellular therapies to ensure their clinical safety and efficacy.

R egenerative medicine has emerged as a promising strategy to restore damaged or diseased cells and tissues as a consequence of aging, disease, injury, or accidents. Typically, these therapies involve deriving cell types of interest from an undifferentiated source of cells with multi- or pluripotency, including human embryonic (hESC) or induced (hiPSC) pluripotent stem cell origin[1]. Following a thorough assessment of specific marker expression, morphology, and functionality of the derived cells in vitro, pre-clinical studies are necessary to assess the survival, integration, safety, and efficacy of the cell product. There are numerous diseases without current curative treatment such as age-related macular degeneration, Parkinson's disease, Type 1 Diabetes Mellitus, and liver disease that could be transformed by the application of cellular therapies to restore tissue function. In fact, currently more than 20 phase I/II stem cell-based studies are registered on ClinicalTrials.gov, but only a few have published results showing survival and safety of the tested cellular therapies; efficacy outcomes are, therefore, awaited (Table 1).

Despite the fact that some cellular therapies have progressed to clinical trials, their immunogenicity remains an unresolved challenge that may impede effective long-term translation to the clinic. Although it is often assumed that autologous hiPSCs (i.e., isogenic grafts) lack immunogenicity, this may be dependent on cell type[2–5]. Alternatively, stem cell-derivatives generated from individuals unrelated to the recipient (i.e., allogeneic grafts) are very likely to lead to immune-mediated rejection due to their allogeneic origin. Different strategies are in place to reduce graft rejection, including the use of immunosuppressants and human leukocyte antigen (HLA) matching between donor and recipient[6,7], but none are entirely successful at abolishing the immune response in a non-toxic manner. Understanding the mechanisms triggering the

[1] Department of Surgery, University of Cambridge, and NIHR Cambridge Biomedical Research Centre, Cambridge, United Kingdom. [2] Peter Gorer Department of Immunobiology, School of Immunology and Microbial Sciences, King's College London, Guy's Hospital, London, United Kingdom. [3] Department of Clinical Neuroscience, University of Cambridge, Cambridge, United Kingdom. [4]These authors contributed equally: Sandra Petrus-Reurer, Marco Romano. ✉email: sp2016@cam.ac.uk; ks10014@cam.ac.uk

**Table 1 Main registered clinical trials with stem-cell-derived products for regenerative medicine.**

| Type of cells | Disease | Format | Clinical trial id (phase) | Type of immunosuppression | Status |
|---|---|---|---|---|---|
| hESC-RPE (MA09-hRPE) | Dry AMD and Stargardt's Macular Dystrophy (SMD) | Suspension (50,000, 100,000 and 200,000 cells) in subretinal space. 13, 13, and 12 participants | NCT01344993/ NCT01345006/ NCT01469832 (Phase I/II) | Low-dose tacrolimus (3–7 ng/mL) and mycophenolate mofetil (range 0.5–2 g orally/day). At week 6 post-infusion only MMF for an additional 6 weeks | Completed[115-117] |
| hESC-RPE (CPCB-RPE1) | Dry AMD (GA) | Parylene scaffold (100,000 cells) in subretinal space. 16 participants | NCT02590692 (Phase I/IIa) | Any current immunosuppressive therapy other than intermittent or low dose corticosteroids | Active, not recruiting |
| hESC-RPE (PF-05206388) | Acute wet AMD | Polyester scaffold (6 × 3 mm) in subretinal space. 2 participants | NCT01691261 (Phase I) | Not indicated | Active, not recruiting |
| hESC-RPE (MA09-hRPE) | Dry AMD | Suspension in subretinal space, (50,000–200,000 cells). 12 participants | NCT01674829 (Phase I/IIa) | Any current immunosuppressive therapy other than intermittent or low dose corticosteroids | Active, not recruiting |
| hESC-RPE (OpRegen) | Dry AMD | Suspension (50,000–200,000 cells), in subretinal space. 24 participants | NCT02286089 (Phase I/IIa) | Not indicated | Active, not recruiting |
| hESC-RPE | Retinitis Pigmentosa | The suspension (150,000 cells) in subretinal space. 10 participants | NCT03944239 (Phase I/II) | Not indicated | Recruiting |
| hESC-RPE | Retinitis Pigmentosa | Therapeutic patch. Subretinal space. 12 participants | NCT03963154 (Phase I) | Mycophenolate Mofetil (MMF) | Recruiting |
| hESC-RPE | Dry AMD | Subretinal space. 10 participants | NCT03046407 (Phase I/II) | Not indicated | Unknown |
| hESC-RPE | AMD, Wet AMD with disciform scar and SMD | hESC-RPE monolayer seeded onto a polymeric substrate versus hESC-RPE suspension in subretinal space. 21 participants | NCT02903576 (Phase I/II) | Not indicated | Completed |
| hESC-RPE (MA09-hRPE) | AMD | The suspension (50,000 cells) in subretinal space. Three participants | NCT01625559 (Phase I) | MMF and Tacrolimus | Completed[118] |
| hiPSC-RPE | Dry AMD (GA) | Biodegradable poly lactic-co-glycolic acid (PGLA) scaffold in subretinal space. 20 participants | NCT04339764 (Phase I/IIa) | Not indicated | Recruiting |
| hESC-RPE | AMD and SMD | Cell suspension in subretinal space. 15 participants | NCT02749734 (Phase I/II) | Not indicated | Unknown |
| hESC-derived Immunity and Matrix Regulatory/ M-cells (CAStem) | Severe COVID-19 | Intravenous infusion of 3, 5, or 10 million cells/kg. 9 participants | NCT04331613 (Phase I) | Not indicated | Recruiting |
| hESC-derived Oligodendrocyte Progenitor cells (GRNOPC1) | Spinal Cord Injury | Injection into the lesion site of 2x10$^6$ cells. Five participants | NCT01217008 (Phase I) | Low dose Tacrolimus | Completed |
| hESC-derived Oligodendrocyte Progenitor cells (AST-OPC1) | Spinal Cord Injury | Injection of 2, 10, 20 × 10$^6$ cells. 25 participants | NCT02302157 (Phase I/II) | Not indicated | Completed |
| hESC-derived Neural Precursor Cells | Parkinson's disease | Cells are stereotactically implanted in the striatum. 50 participants | NCT03119636 (Phase I/II) | Not indicated | Unknown |
| hESC-Astrocytes (AstroRx) | Amyotrophic Lateral Sclerosis (ALS) | Intrathecal (spinal) injection of 100, 250, 2 × 250, and 500 × 10$^6$ cells. 21 participants | NCT03482050 (Phase I/II) | | Completed |
| hESC-Progenitors (CD15$^+$Isl-1$^+$) | Severe Heart Failure | Fibrin patch embedded hESC-progenitors. Ten participants | NCT02057900 (Phase I) | Cyclosporine and Mycophenolate Mofetil | Completed[119] |
| hESC-Pancreatic Precursor cells (VC-02) | Type 1 Diabetes Mellitus (T1DM) | Cells loaded into a delivery device subcutaneously. 6 participants | NCT03162926 (Phase I) | Not indicated | Completed |
| hESC-Pancreatic Precursor cells (VC-01/VC-02) | Type 1 Diabetes Mellitus (T1DM) and T1DM with Hypoglycemia | Cells loaded into an encapsulation device implanted subcutaneously. 75 and 69 participants | NCT03163511; NCT02239354 (Phase I/II) | Not indicated | Recruiting; active not recruiting |
| Allogenic hiPSC-Cardiomyocytes | Heart failure | 100 × 10$^6$ cells in 2.5–5 mL medium suspension injected into the myocardium. 5 participants | NCT03763136 (Phase NA) | Not indicated | Recruiting |
| hiPSC-Myocardium | Terminal heart failure | Implantation of engineered heart muscle (hiPSC-cardiomyocytes and stromal cells in bovine collagen type I hydrogel) on the dysfunctional left or right ventricular myocardium. 53 participants | NCT04396899 (Phase I/II) | Not indicated | Recruiting |

immune response toward a cellular product remains key to minimizing rejection and ensuring long-term graft survival and function.

## What are the dominant drivers of the immune response to stem cell therapies?

The immune response is orchestrated by the innate and adaptive immune systems. The first line of defense or the innate immune response consists of fast (0–96 h) and usually non-specific physical, chemical, and cellular responses against pathogens. The second line of defense or the adaptive immune response also referred to as acquired immunity, is only found in vertebrates and is long-lasting. Although innate and adaptive immune responses are often considered as sequential and separate, there is now increasing recognition that there is an overlap between the two, both mechanistically and in timelines (Table 2).

Many of the pathways and processes in innate immunity are the same as those responsible for non-specific reactions to tissue damage, resulting in inflammation. Therefore, tissue stress and inflammation that can occur at the time of transplantation of regenerative cellular therapies is predominantly mediated by the innate immune system.

Natural killer cells (NKs), a key component of the innate immune system, are thought to play an important role in allogeneic cellular product rejection (Fig. 1 and Table 2). NKs specifically target and kill cells that express mismatched/lack self-human leukocyte antigen class I (HLA-I, also known as major histocompatibility complex [MHC]) molecules ("missing-self

**Table 2 Main components, dominant stimuli, and effector functions of the innate and adaptive immune response to regenerative cellular therapies.**

| | Main component | Dominant stimuli | Effector functions |
|---|---|---|---|
| **Innate immune system**<br><br>*Fast (0–96 h) and usually non-specific physical, chemical, and cellular responses against pathogens* | PAMP and DAMP<br>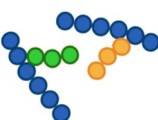 | – Damage-associated molecular patterns (DAMP) and Pathogen-associated molecular patterns (PAMP) via pattern recognition receptors (PRR)[8] | – Immune cell recruitment<br>– Cytokine and chemokine release<br>– Inflammation<br>– Adaptive immunity<br>– Tissue repair |
| | NK Cell<br>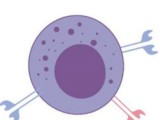 | – NK cell activated by lack of self HLA class-I (HLA-I), mediated by inhibitory and activating molecules including DNAX Accessory Molecule-1 (DNAM-1) receptor and killer-cell immunoglobulin-like receptors (KIR)[9,10] | – Direct lysis and cytotoxicity through perforin, granzymes, and tumor necrosis factor (TNF) family effector molecules (Fig. 1)<br>– Release of pro-inflammatory cytokines<br>– Recognition and killing of target cells opsonized with antibodies via low-affinity IgG receptor CD16 receptor |
| | Complement and coagulation system<br>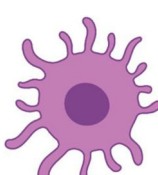 | – Direct activation by pathogens or indirectly by pathogen-bound antibodies | – Membrane attack by rupturing cell wall (classical complement pathway)<br>– Phagocytosis by opsonizing antigens (alternative complement pathway)<br>– Inflammation by attracting macrophages and neutrophils (lectin pathway) |
| | Dendritic Cell<br>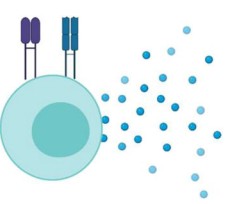 | – Direct by recognition of alloantigens/pathogen molecules (PAMP and DAMP)<br>– Indirectly by inflammatory mediators and cytokines | – Activation of CD4+ T helper cells and the innate immune system |
| **Adaptive immune system**<br><br>*Antigen-specific and reacts to a broad range of microbial and non-microbial foreign substances or antigens; quick and vigorous response with repeated exposure to the same antigen (memory)* | CD4+ T Helper Cell<br>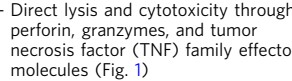<br><br>Three signals for T cell activation[34]:<br>Interaction between HLA and the T cell receptor (TCR)<br>Provided by APC expression of co-stimulatory molecules B7.1 [CD80] or B7.2 [CD86]<br>Release of specific cytokines, which also determines CD4+ T cell commitment towards different T-helper subsets | – Recognition of foreign/mismatched donor antigens via alloantigen-HLA-II-TCR complex[34,35] | – Direct, semi-direct or indirect pathways of allorecognition[37]: activation of other immune cells including cytotoxic T cells, B cells, and NK cells (Fig. 2) |
| | CD8+ T Cytotoxic Cell<br>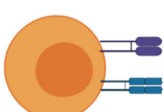<br><br>*Up-regulation of CTLA-4 or PD-1 contribute to inhibition of T cell activation[135,140] | – Recognition of foreign/mismatched donor antigens via alloantigen HLA-I-TCR complex (usually on nucleated cells)[34] | – Direct cytotoxic lysis: allogeneic graft clearance |

**Table 2 (continued)**

| | Main component | Dominant stimuli | Effector functions |
|---|---|---|---|
| 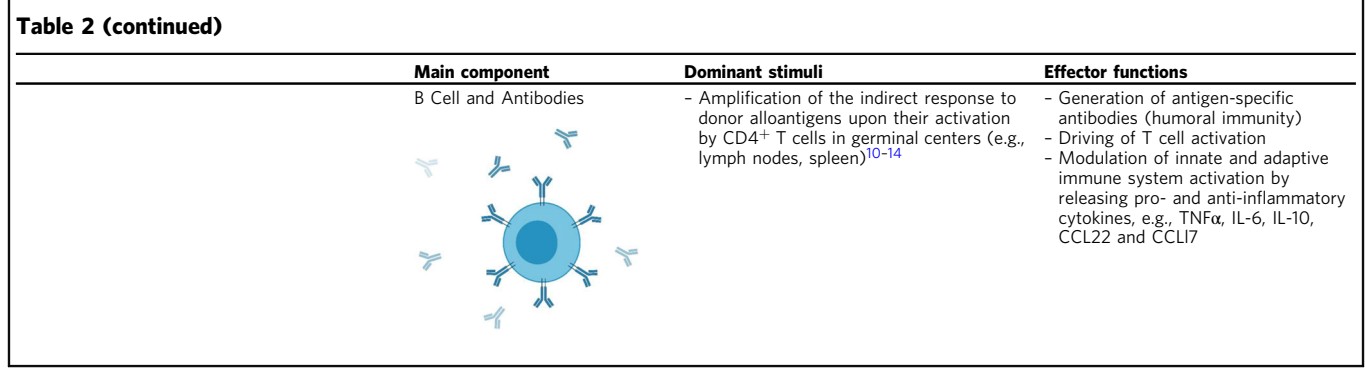 | B Cell and Antibodies | – Amplification of the indirect response to donor alloantigens upon their activation by CD4+ T cells in germinal centers (e.g., lymph nodes, spleen)[10-14] | – Generation of antigen-specific antibodies (humoral immunity)<br>– Driving of T cell activation<br>– Modulation of innate and adaptive immune system activation by releasing pro- and anti-inflammatory cytokines, e.g., TNFα, IL-6, IL-10, CCL22 and CCLl7 |

hypothesis)[15,16], a response that will also trigger additional adaptive immune responses via specific cytokines leading to clearance of donor graft cells. NK-mediated killing may be aggravated by specific culture conditions used during the generation of stem cell therapies, which might give rise to differentiated cellular products improperly expressing immune molecules or containing reminiscent molecules of the cell type of origin[17–22] (see "What are the immunological confounders that should be taken into consideration for stem cell therapy?"). Conversely, certain tissues might, either naturally or due to the inflammatory setting generated at the time of transplantation, express inhibitory ligands or cytokines at high levels which might compensate for the degree of HLA-mismatch. This has been shown for HLA-C, HLA-E up-regulation or IL-10 secretion in hPSC-retinal pigment epithelial (RPE) cells[23–25]. In fact, recent works have demonstrated that the incorporation of specific NK inhibitory ligands (e.g., HLA-E, CD47) into the cellular therapies can ameliorate NK cell graft response[26–28] (see "How can immunogenicity be overcome or ameliorated for the long-term success of cellular therapies?").

The complement system, composed of more than 30 soluble and cell-bound proteins[29], is another major component of the innate immune system. Its relevance is illustrated by the finding that the success of islet and hepatocyte cell transplants is limited by activation of the complement and coagulation cascades[30,31]. Also, hiPSC-RPE cells have shown to produce several complement components which, under inflammatory or stress conditions, had a significant impact on RPE cell survival and function in a therapeutic setting[32,33].

In the context of transplantation, adaptive immunity is triggered by the recognition of foreign/donor antigens by recipient T cells. This is mostly due to the expression of highly polymorphic MHC molecules leading to T cell-mediated immune response and rejection of the transplanted allogeneic cells[34,35] (Table 2). In general, HLA-I-peptide complexes (expressed on a majority of nucleated cells) are recognized by CD8+ cytotoxic T cells and following activation they mediate the direct killing of the target cells; whereas HLA-II molecules expressed mainly on antigen presenting cells (APC) present peptide derived from exogenous antigens to CD4+ helper T cells, leading to the activation of other immune cells including cytotoxic T cells and B cells[36]. Transplanted cells can activate the adaptive immune system (alloreactive T cells) through three pathways of allorecognition[34,37] (Fig. 2). In the direct pathway of allorecognition, recipient CD8+ and CD4+ T cells recognize HLA-I or II-allo-peptide complexes, respectively, on donor APCs/cellular therapy which are consequently activated. In the indirect pathway of allorecognition, donor allo-peptide is processed by recipient APCs and presented to mainly CD4+ T cells. In the semi-direct pathway, intact donor HLA-I and -II-allo-peptide complexes are internalized, transferred to the membrane of recipient APCs, and

presented to CD8+ or CD4+ T cells for their activation[38]. Activated recipient CD4+ T cells (with direct or indirect specificity) can provide help to cytotoxic CD8+ T cells with direct allospecificity to kill donor cells by recognizing allo-HLA-I molecules on their surface (typically leading to acute rejection); in addition to activating other immune cells e.g., NK cells and B cells which in turn will produce allograft-specific antibodies (typically leading to chronic rejection)[39,40]. In fact, different degrees of mismatch would be anticipated to result in different rejection strengths, as has been shown by Sugita and colleagues when allogeneic hiPSC-RPE cells with homozygous HLA-A, -B, -DRB1 alleles greatly reduced both in vitro and in vivo immune responses[41,42].

Because most regenerative cellular therapies are not expected to contain donor HLA-expressing APCs, it is anticipated that indirect and semi-direct allorecognition will dominate the adaptive immune response to regenerative cellular therapies. However, remnant embryonic antigens such as TRA-1, SSEA3 expressed by cellular therapies (even in an isograft setting) with direct T cell activating capability may be able to elicit an immune response. Therefore, although certain cellular products would be expected to activate the adaptive immunity directly, the majority would be expected to activate the indirect and semi-direct mechanisms of allorecognition through recipient dendritic cells (DC) or macrophages that would recognize either alive or dead donor grafted cells and present their respective allo-antigens to T cells (via processing and presentation of donor antigens or following the transfer of intact donor HLA-allo-peptide complexes)[38]. Nonetheless the up-regulation of certain inhibitory T cell ligands (e.g., PD-L1, CD95), the secretion of specific molecules such as pigment epithelium-derived factor (PEDF), IL-10, TGFβ, or the induction of alloantigen-specific regulatory T cells (Tregs), that may be inherent to the immunosuppressive phenotype of a specific cell type or occur under specific inflammatory conditions and cytokines (e.g., IFN-γ), could favor the inactivation of T cells even in the presence of a degree of HLA-mismatch[23,25,41–46]. Taking advantage of this, Yoshihara et al. elegantly over-expressed PD-L1 and showed long-term survival of the injected edited human islet-like xenografts. These xenografts restored glucose homeostasis in immune-competent diabetic mice for 50 days, and upon ex vivo IFN-γ stimulation, they showed restricted T cell activation and graft rejection compared to non-engineered cells[47].

Conversely, the presence of inflammatory molecules (particularly IFN-γ) may induce the up-regulation of HLA-II molecules in the cellular therapies, which would mainly trigger the direct pathway of allograft rejection via CD4+ cells leading to B cell activation, as shown by multiple studies in hPSC-RPE[23–25,41,42,45]. Despite the intense historical interest in studying the impact of knocking out HLA genes on immunogenicity, recent advances in genetic engineering tools have finally provided a robust platform for several groups to very attractively generate cells lacking HLA genes specifically aimed at developing universal

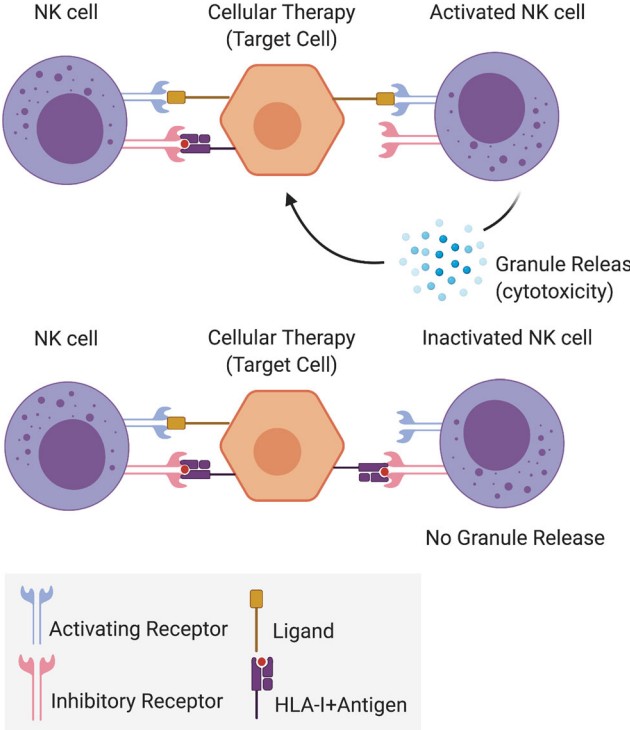

**Fig. 1 The innate Immune system.** NK cell activation or inactivation following activating or inhibiting receptor-target cell ligand signaling. A mismatched/lack of HLA-I-Antigen complex is a strong NK cell-activating signal. Once activated, a cytotoxic response will follow mainly through granule release. HLA: human leukocyte antigen, NK: natural killer.

cellular therapies for regenerative medicine[23,48]. However, this strategy particularly intended at abolishing T-cell responses may eventually lead to NK cell activation and graft rejection, hence opening a novel window for future therapeutic strategies aiming to keep both adaptive and innate immune systems under control (see "How can immunogenicity be overcome or ameliorated for the long-term success of cellular therapies?").

**What are the available experimental platforms to study the immunogenicity of cellular therapies in vitro and in vivo?**
An increasing broad range of in vitro and in vivo experimental approaches are available to investigate the immune response to cellular therapies. Due to the complexity of the immune system and the inherent deficiencies of all methodologies, it is almost always necessary to combine data from several in vitro techniques and in vivo models in order to reach definitive conclusions on the immunogenicity of cellular therapy (Table 3). However, it is not usually feasible for a research group to incorporate the complete range of in vitro and in vivo methodologies into every study. Thus, the assessment of the response of a particular immune compartment is often the focus of individual studies. Here we describe a summary of the most utilized methods to comprehensively evaluate the immunogenicity of cellular therapies.

**In vitro methodologies to assess the immunogenicity of cellular therapies.**

*Assessment of innate immunity*
NK responses to a cellular therapy can be assessed by measurement of NK activation or degranulation (i.e., lytic granule trafficking), cytokine release, and NK cytotoxic killing. NK cell degranulation assays are usually performed

by co-culturing human pre-activated NK cells and target cells (cellular therapy) in an assay-dependent ratio. Degranulation is typically evaluated by the expression of CD107 (lysosomal-associated membrane protein 1/LAMP-1) molecule and intracellular cytokines (e.g., IFN-γ) in different subsets of NK cells, which can be co-labeled with specific KIR-ligands. Cytotoxic activity of NK cells towards a target cell can be measured in vitro by quantifying the release of naturally occurring substances (e.g., lactate dehydrogenase), or through the use of target cells labeled with radioactive compounds, such as $^{51}$Chromium- or $^{111}$Indium, which are released upon NK cell killing[49,50]. Alternative methods to measure NK cell cytotoxicity include flow cytometry (e.g., intracellular levels of perforin, granzymes, and granulysin), enzyme-linked immunosorbent assay (ELISA)-based granzyme measurement, image cytometry, and morphometric analysis by microscopy[51,52]. Additionally, NK cell-mediated cytolysis and antibody-dependent cellular toxicity (ADCC) can also be studied in a novel label-free, non-invasive manner using image cytometry or a microelectronic sensor measuring impedance in adherent cells, thus avoiding the use of radioisotopes[53,54]. Despite these novel methods, most studies assessing NK activation by PSC aimed as genetically engineered-hypoimmunogenic cellular therapies used $^{51}$Chromium release assays, which illustrated the decrease in NK cell cytotoxic levels of the respective gene editing approaches[26–28,55] (see section "How can immunogenicity be overcome or ameliorated for the long-term success of cellular therapies?").

To complement the evaluation of the innate immune system, some studies have assessed by flow cytometry or immunofluorescence (IF) staining the expression of activating or inhibitory ligand molecules that might be expressed by the hPSC-derived cells. These, for instance, include HLA-A, B, C, CD112, CD155, PCNA, and NKG2A shown by Petrus-Reurer et al. in hESC-RPE cells[23]; or NKG2D, NKp80, NKp16, NKp44, and NKp30 shown by Deuse et al. in hiPSCs, hi-endothelial-like cells and hi-cardiomyocyte-like cells[26]. Although no conclusive ligand expression differences were found between edited and respective wild-type counterparts in these studies, their unique expression and combined downstream signal interaction might explain some of the mechanisms driving NK cell activity in specific cell lineages[57,58].

Assessment of other components of the innate immune response includes analysis of pattern recognition receptors (PRR) by qPCR before and after stimulation with pro-inflammatory cytokines as shown by Fischer et al. with human iPSC-Hepatocytes[59]. However, the majority of the studies regarding PRRs have so far been aimed to define a better infectious disease model rather than defining the contribution of PRRs in cell rejection. Key components of the complement cascade (Collectin-11, MASP2, C3d, C5B-9 (mac), CFB) have been analyzed before and after cell stress or pre-conditioning with TNF-α or IFN-γ[32,33]. A pioneering study from Sugita et al. showed by using different assays (flow cytometry, RT-PCR, and ELISA) that iPSC-RPEs expressed key components of the complement cascade that are up-regulated by IFN-γ and T Helper-1 mediated responses[32]. A few years later, Fanelli et al. showed by immunohistochemistry (IHC) and western blotting that in vitro cultured hiPSC-RPE under hypoxic stress upregulated their surface expression and the release of collectin-11, a molecule triggering complement activation[33].

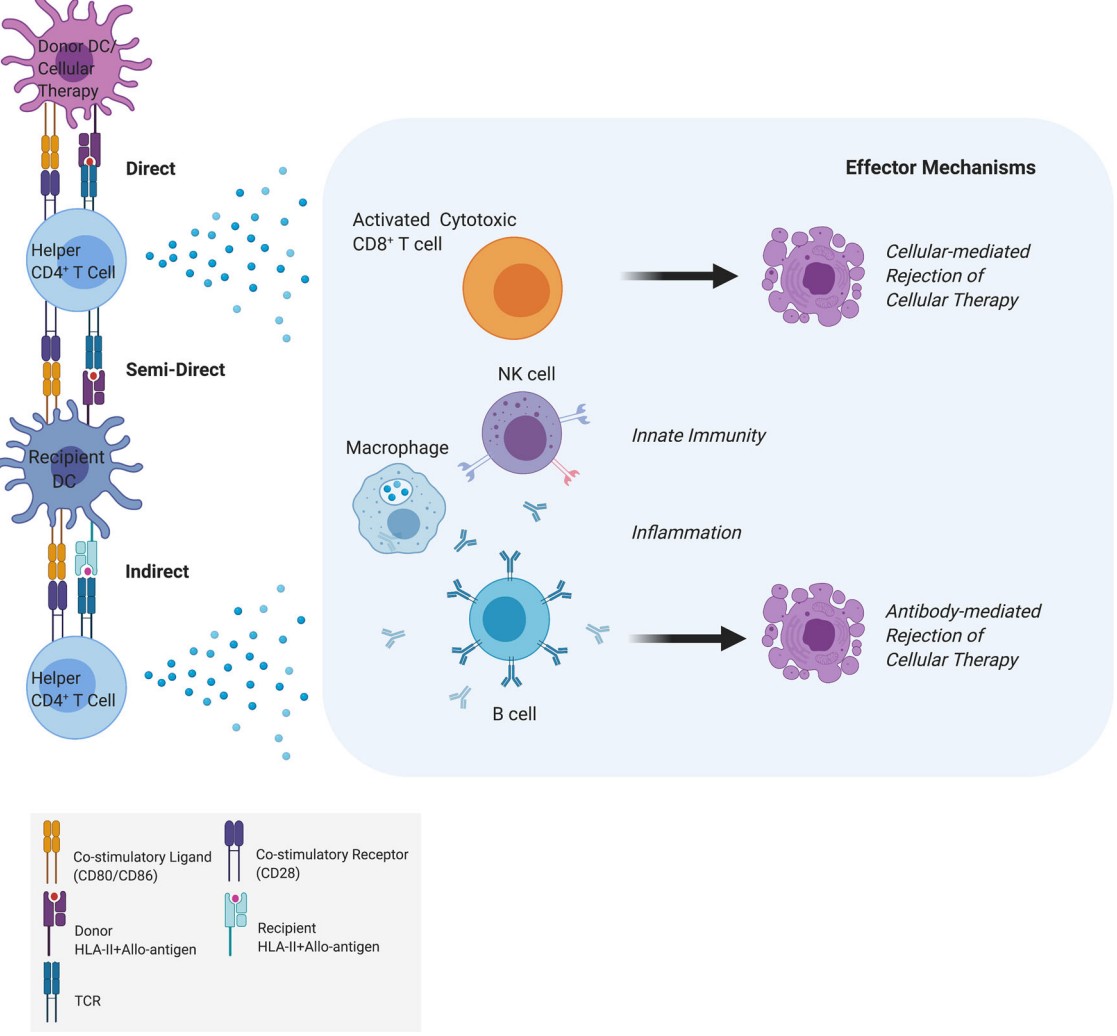

**Fig. 2 The adaptive immune system.** Illustration representing the three different pathways of allorecognition. The *direct pathway* of allorecognition typically involves recognition of intact HLA-I or -II-Antigen complexes expressed by donor DC (i.e., APC)/cellular therapies by recipient CD4+ or CD8+ cells, respectively, usually leading to acute graft rejection. T cells with direct allospecificity are present in all individuals at very high frequency and this pathway is thought to play a major role immediately following transplantation. The *indirect pathway* of allorecognition involves processing and presentation of donor HLA molecules by recipient DC (i.e., APC) to recipient CD4+ T cells, which then provide help for CD8+ T cell-mediated cytotoxic killing and antibody production by B cells. The frequency of T cells with indirect allospecificity is undetectable but increases with time from the transplant. In line with this, this pathway was thought to be the most relevant for graft rejection late post-transplant. The *semi-direct pathway* involves the transfer of intact donor-derived HLA molecules to recipient APC leading to CD8+ or CD4+ T cell activation. This latter pathway implies that the direct pathway of allorecognition lasts for longer than what was initially thought and indicates that the same recipient DC can present directly and indirectly donor HLA molecules to host T cells. In all pathways, the activated recipient CD4+ T cells provide help for activation of cytotoxic CD8+ T cells which kill donor cells by binding to allo-HLA-I on their surface then leading to cellular-mediated rejection of cellular therapy (typically acute reaction). In addition, activated CD4+ T cells will trigger the innate immune system, inflammation, and B cell maturation into plasma cells that will produce allo-antigen specific antibodies which will lead to an antibody-mediated rejection of the cellular therapy (typically chronic rejection). Cellular therapy: refers to an HLA-II expressing target cell, DC: dendritic cell, TCR: T cell receptor, HLA: human leukocyte antigen, NK: natural killer.

### Assessment of cell-mediated T cell activation

The most commonly used in vitro methodology to study the response of T cells with direct allospecificity to a specific cellular therapy is a co-culture system with the cells of interest and allogeneic peripheral blood mononuclear cells (PBMCs). To try and mimic the indirect alloresponse, monocyte-derived DCs can be co-cultured with autologous T cells in the presence of the cellular product[61]. However, the indirect (and semi-direct) allorecognition is best modeled in vivo as it takes time for this pathway to emerge after antigen exposure. Experimental end-points for these co-culture assays comprise immune cell proliferation, expression of activation markers, target cell killing, and cytokine production. These assays are usually complemented with the evaluation of the expression of HLA and co-stimulatory and co-inhibitory molecules on the cellular product (Table 3). T cell proliferation measured in these assays can be coupled to the evaluation of naïve, activated, central and effector memory T cell sub-populations using specific markers[62], thus providing a more comprehensive evaluation of the target T cells activated by the cellular product[25,40,42,63–68]. To understand how the inflammatory microenvironment influences a cellular therapy and its capacity to activate T cells, co-cultures can be executed

**Table 3 Summary of current immunological methodologies to assess the immunogenicity of cellular products in vitro and in vivo.**

| Immune compartment | | | Immunological assay — In vitro | Immunological assay — In vivo |
|---|---|---|---|---|
| Innate immunity | | | • NK cell degranulation/activation (CD107 flow cytometric analysis) and cytotoxicity ($^{51}Cr/^{111}In$ release, impedance measurements)[23,26–28,55,56] <br> • Evaluation of inhibitory or activation ligands by flow cytometric/immunofluorescence analysis in grafted cells (e.g., HLA-ABC, HLA-C, HLA-E, NKG2D, MIC-A/B, CD155, CD112, PCNA, CD47, CD55, CD59)[20,23,56] <br> • Complement cascade, DAMP secretome[33,60] | • Immunohistochemical/flow cytometric quantification of graft immune phenotype and infiltration with NK cells (CD56+), neutrophils (CD11b+, CD66b+, CD33+), macrophages (RAM11+)[23] <br> • Deposition of complement molecules in the graft (e.g., CL-11, C3d, C4b-9), collectin-11, DAMP secretome[32] <br> • Complement molecule concentration in serum (e.g., C3a, C5a, CFB, MAC, CFH, CFI)[60] |
| Adaptive Immunity | Humoral | B Cell Activation | • No available assays at present | • Flow cytometric characterization of transitional, naïve, memory and plasma cell phenotypes (e.g., CD10, CD19, CD20, CD23, CD27, CD38, IgM, IgG)[40,167] <br> • Ex vivo B cell ELIOSPOT assays[167] <br> • Splenic germinal center formation[167] |
| | | Antibody Production | • No available assays at present | • Total serum human IgG and IgM levels <br> • Incubation of serum from rejecting animals with donor cells followed by fluorescently-labeled anti-human IgG or/and IgM staining for flow cytometric analysis[23,40] <br> • IgG and B immunohistochemistry analysis[32] <br> • Production of HLA-specific anti-donor antibodies with Luminex assay[32] |
| | Cellular | CD8+ T Cells | • Co-culture with cellular product and characterization of T cell proliferation (CFSE-labeled T cells; production of IFN-γ) and the main CD8+ T cell subsets by flow cytometry[23,25,40–42,45,46,55,62,64,65] <br> • Cytotoxic activity by $^{51}Cr/^{111}In$ release assay[55] <br> • Ex vivo activation in Granzyme B ELISPOT assay | • Immunohistochemical/flow cytometric quantification of graft immune phenotype and infiltration with lymphocytes (CD3+, CD8+)[4,5,23,39–41,46,64,68] |
| | | CD4+ T Cells — Common | • Co-culture with the cellular product and characterization of T cell proliferation (CFSE-labeled T cells; production of IFN-γ)[74] and the main CD4+ T cell subsets by flow cytometry[23,25,40–42,45,46,55,62–68] | • Immunohistochemical/flow cytometric quantification of graft immune phenotype and infiltration with lymphocytes (CD3+, CD4+)[4,5,23,40,41,46,66,83] |
| | | CD4+ T Cells — Indirect Pathway | • Ex vivo proliferation (CFSE-labeled T cells; production of IFN-γ in ELIOSPOT assays) in response to DCs from the donor used for reconstitution of the immune response/ recipient pulsed with preparations of the grafted cells[46] | |
| | | CD4+ T Cells — Direct Pathway | • Ex vivo proliferation (CFSE-labeled T cells; production of IFN-γ in ELIOSPOT assays) in response to DCs from the donor used for generation of grafted cells (or third-party control DCs) cells with/without specific cytokines and ligand/antibody-specific antibodies[46] | |
| Common | | | • Evaluation of cytokine production/proliferation from in vitro co-cultures, sera/tissue from transplanted cells (e.g., Multiplex Luminex of IFN-γ, IL-1b, IL-6, IL-17 or by tissue immunofluorescence)[23,25,39–42,45,46,55] | Multiplex Luminex of IFN-γ, IL-1b, IL-6, IL-17 or by tissue immunofluorescence production[63,67,73,74] |
| Cellular products | | | • Evaluation of the of inhibitory or activation ligands by flow cytometric/immunofluorescence analysis of cellular product (e.g., CD80, CD86, PD-L1, PD-L2)[23,25,41,42,48,64,65,67,74] and evaluation of anti and pro-inflammatory cytokine production[63,67,73,74] | CD80, CD86, PD-L1, |

following pre-conditioning of the hPSC-derived product with pro-inflammatory cytokines such as IFN-γ, TNF-α, and IL-1β (Table 3). Generally, co-cultures of the cellular therapies with PBMCs or isolated CD4[+] or CD8[+] tracker-labeled (e.g., CFSE) to assess T cell proliferation or IFN-γ measurements in the supernatants to assess T cell activation has been the most utilized approach to evaluate the immunogenicity of cellular therapies from matched, mis-matched or genetically modified donor cells[23,40–42,45,48,55]. And in fact, most cellular products tested in this manner have shown little or no immunogenicity, thus confirming their inability to directly activate the adaptive immune system. Specifically, by co-culturing PBMCs with hESC-RPEs pre-stimulated with IFN-γ, Petrus-Reurer et al. have recently shown their lack of immunogenicity unless cultured with very specific conditions (e.g., particular target:immune cell ratio, presence of cytokines and activating molecules). In a similar manner, this lack of immunogenicity in vitro has also been shown for other cell types like hiPSC-derived cartilages[69]. Recently, Mehler et al. showed that hiPSC-derived neural crest stem (NCS) cells induced negligible CD3[+] T cell proliferation although, conversely, hiPSC-derived smooth muscle cells (SMC) favored a high level of cell proliferation[65].

*Assessment of the immunomodulatory function of the cellular therapy*

Indeed, in contrast to immune activation, suppression of T-cell responses in vitro has been described for several cellular products (Table 3). Typically, this 'immunomodulatory' potential is assessed by quantifying the impact of the cellular therapy on T cell proliferation in the presence of molecules that would induce it (e.g., soluble antibodies anti-CD3/OKT-3, coated beads, IL-2)[23]. Assessment of expression of co-stimulatory molecules (inhibitory: e.g., PD-L1, PD-L2; activating: e.g., CD80, CD86) by the cellular therapy as shown by several studies is also relevant, as this may confer T cell inhibitory properties and explain some of the inhibition of T cell proliferation and cytokine production observed in the co-culture in vitro assays[63,70–73]. Petrus-Reurer et al. have shown that pre-IFN-γ stimulated hESC-RPEs significantly suppressed both CD8[+] and CD4[+] T cell proliferation in the presence of OKT-3, a mechanism that might be mediated by the up-regulation of specific co-inhibitory molecules in the absence of co-stimulatory molecules. Similarly, hPSC-derived retinal ganglion cells had the ability to inhibit T cell proliferation in a transforming growth factor (TGF)-β2 dependent way[74]. TGF-β production by hiPSC-NCS is crucial for mediating their T cell proliferation immunosuppressive capacity as shown by Fujii et al[67]. However, by using different experimental conditions (e.g., source of hiPSC, NCS differentiation protocol, length and readout of the MLR), Mehler et al. showed the lack of this capacity for hiPSC-NCS cells. The opposite results obtained in these two studies highlight the importance of the immunological confounders present at the time of performing the different in vitro and in vivo assays (see "What are the immunological confounders that should be taken into consideration for stem cell therapy?").

**In vivo methodologies to assess the immunogenicity of hPSC-derivatives**. Availability of optimized and representative in vivo animal models are a limiting factor for the definitive study of the immunogenicity of regenerative cellular therapies. Human cellular products have been previously transplanted into different anatomical locations using xeno- and immunodeficient animal models (e.g., rodents, lagomorphs, pigs, non-human primates)[23,41,75,76]. Despite species differences, these studies have been very useful in understanding the basic mechanisms of the host response. However, as these models do not replicate the human immune response to human cellular therapies, their ultimate utility for clinical translation is somewhat limited and they will not be further discussed in this review.

There are a number of effective strategies to model a competent human immune compartment in vivo in order to assess the immunogenicity of cellular therapies[77–80]. In essence, these models involve the reconstitution of severely immunodeficient rodents with human immune cells. The human immune cells used for reconstitution of the mice can either be in the form of adoptively transferred mature lymphocytes (PBMCs) or generated de novo by transfer of CD34[+] hematopoietic cells. Mature lymphocytes are usually obtained from the peripheral blood of human donors, although the spleen of deceased transplant organ donors has recently been identified as an alternative source[81]. Alternatively, CD34[+] HSCs can be obtained from umbilical cord blood, bone marrow, or fetal liver. The most common strain of mouse used for the generation of these "humanized" or human immune system (HIS) mice is NOD scid gamma (NSG) mice which lack mature B, T, and NK cells. It is important to note that the immune compartment generated in HIS mouse models is "suboptimal", and a number of different strategies have emerged to augment and improve the immune response mounted by the animals. These include co-transplantation of fetal human thymic or other tissue and the use of genetically modified mice expressing human cytokines, HLA, or other important signaling molecules. The range of available humanized mouse models and their relative advantages and disadvantages has been reviewed in multiple studies[77–80], and are also summarized in Fig. 3.

A further challenge is the occurrence of graft-versus-host (GvH) reaction which limits the experimental time-window[82]. This window is dependent on the HIS model and mouse strain used and is most limited in the PBMC model. Critical in interpreting the observed immune response is the use of positive and negative control tissues/cells. Positive control cells should ideally be "adult" cells and tissues that may represent an alternative treatment to cellular therapy. Where possible, autologous cells/tissues are the ideal negative controls although practically this is rarely possible. Lastly, the HIS models are subject to significant biological variation and large group sizes may be necessary to draw definitive conclusions. However, the generation of HIS mice from the spleen of deceased transplant organ donors, together with the use of control cells and generation of cellular therapies from the same donor, has the potential to address some of these challenges[81].

The in vivo immunogenicity of hPSC-cell grafts can be quantified using a range of experimental endpoints, which have been summarized below from a variety of studies evaluating stem cell therapies (Table 3):

- Assessment of survival/immune destruction of the transplanted cells by luciferase, specific human- or tissue-specific markers on IHC/immunofluorescence (IF)[23,26–28,39,40,55,66].
- Immune characterization of the transplanted cells (e.g., upregulation of immune markers such as HLA-I and -II, or the above-mentioned T cell and NK cell activating and inhibitory ligands) by flow cytometry or IHC/IF[23,28,41].
- Phenotypic characterization of the immune infiltration (type and amount), e.g., CD3[+]/CD4[+]/CD8[+] T cells[4,5,64,66,68,83]; CD56[+] NK cells; RAM11[+] macrophages; HLA-II[+] APCs by

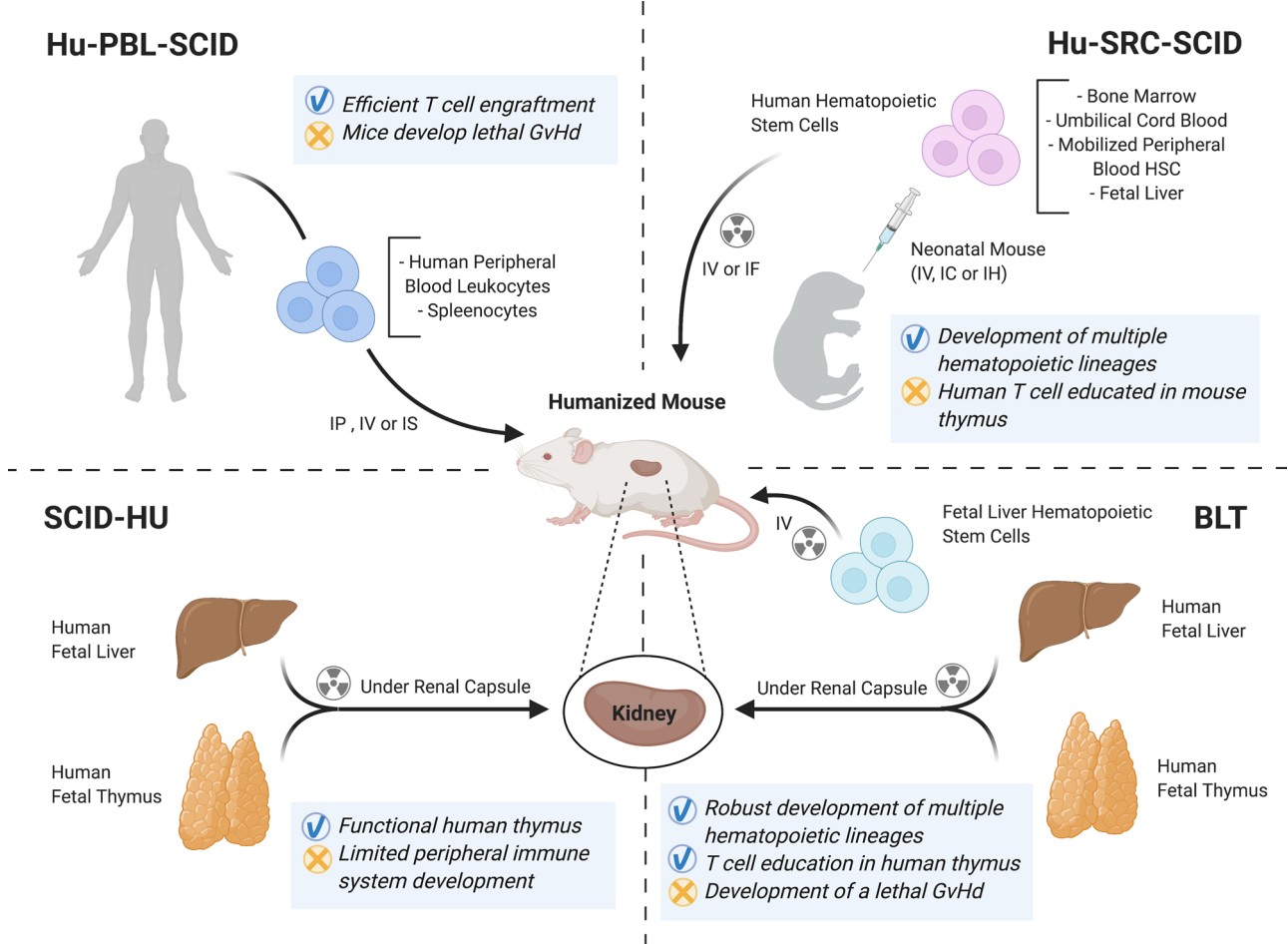

**Fig. 3 Humanized mice models for the in vivo immunological assessment of cellular therapies.** Illustration representing the methodologies behind current humanized mice models highlighting the advantages (check) and disadvantages (cross) of each model. The models include: the human peripheral blood lymphocytes (*Hu-PBL-SCID*) in which most of the engrafting cells are human T cells that express an activated phenotype while few B cells or myeloid cells engraft. One caveat is that these mice will develop a xenogeneic graft-versus-host disease (xeno-GVHD) that results in death, but xeno-GVHD can be delayed using immunodeficient mice lacking mouse MHC class I or class II; the human stem cell repopulating cell (*Hu-SRC-SCID*), which is established by engraftment of human hematopoietic stem cells (HSC) derived from bone marrow, umbilical cord blood, fetal liver, or mobilized peripheral blood HSC. Engrafting mature adult immunodeficient IL2rγ null mice with HSC permits the generation of multiple hematopoietic cell lineages but few T cells while human T cells are readily generated following engraftment of newborn or 3–4 week-old NSG and NOG mice with HSC; the *SCID-HU*, which is established by implantation of human fetal liver and thymus fragments under the renal capsule of immunodeficient mice and a major limitation is the paucity of human hematopoietic and immune cells in the peripheral tissues; and the bone marrow, liver, thymus (*BLT*), which is established by implantation of human fetal liver and thymus fragments under the renal capsule of sublethally irradiated immunodeficient mice accompanied by intravenous injection of autologous fetal liver HSC. The use of immunodeficient NOD-scid mice to establish the BLT model led to human immune system engrafted mice, which is further enhanced by the engraftment of NSG mice. A complete hematopoietic and immune system develops, and the human T cells are educated on a human thymus and are HLA-restricted. IP: intraperitoneal, IV: intravenous, IS: intrasplenic, IF: intrafemoral, IC: intracardiac, IH: intrahepatic, GvHd: graft versus host disease.

flow cytometry or IHC/IF to evaluate the immune responses elicited by the graft[23,28,39,66].

- Systemic cellular immune response (e.g., increase in the number of lymphocytes in spleen) or change in phenotype assessed by flow cytometry and IHC/IF for specific T cell and NK cell populations, but also assessment of splenic germinal center formation for B cells or deposition of complement molecules[14], DAMP secretome by IHC/IF[66].
- Systemic anti-human antibody-mediated response assessed by flow cytometry or Luminex assays of the transplanted mice sera[23,32,40].
- Evaluation of the main complement molecules (e.g., C3, C5, CFH/B/I, CL-11, C3d, and C5b-9) present in the host serum or deposited at the site of the graft, measured by ELISA and/or IHC[32,33].

The combination of all the described assays would be the ideal approach to get the full picture of the immune response induced in vivo, but often it is not possible and only some of the mentioned endpoints are analyzed. As an exemplar study, Petrus-Reurer et al. evaluated the immunogenicity of unedited wild-type and edited hESC-RPE lacking HLA-I and -II molecules injected in the subretinal space of albino rabbits. They assessed the survival of engrafted human cells, immune marker expression (HLA-I, HLA-II), and immune response (CD3, CD56, RAM11) with IF. Furthermore, they evaluated specific anti-human antibody production by incubation of rabbit sera that received the cellular therapies with hESC-RPE ex vivo further mixed with anti-rabbit fluorescent antibodies analyzed by flow cytometry. Overall, this study demonstrated decreased early and delayed late immune responses of HLA-I

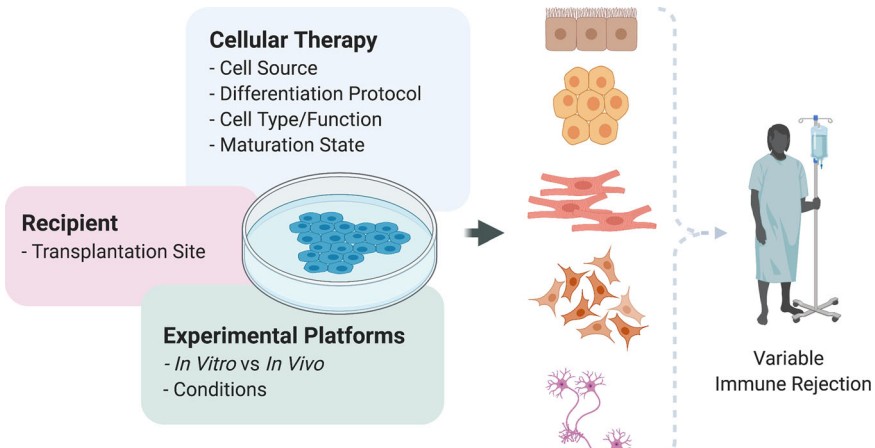

**Fig. 4 Immunological confounders of cellular therapies.** Illustration representing the immunological confounders involved in cellular therapies including factors concerning various aspects of the cell therapy (e.g., cell source, differentiation protocol, cell type/function, specific maturation state of the cell product), the recipient's transplantation site, and the limitations inherent to in vitro and in vivo experimental platforms. Overall these factors might trigger different degrees of rejection in the patient that will receive the cellular therapy, therefore limiting its efficacy.

and -II knockout hESC-RPE compared to wild-type cells in a xenograft model without immunesuppression[23].

These assessments can also be complemented by the use of real-time imaging techniques (e.g., using luciferase constructs, spectral domain-optical coherence tomography [SD-OCT] scans in the retina) or behavioral assays for dopaminergic cells as alternative methods to assess immune infiltration, graft cell survival, and/or function of the cellular therapy in the site of injection. This has been elegantly shown by several groups exploring survival and function of e.g., hypoimmunogenic hiPSC/hESC-RPE cells to be used as an alternative treatment modality for age-related macular degeneration[23,26,28].

### What are the immunological confounders that should be taken into consideration for stem cell therapy?

There are several confounders, both cell-related and assay-related, that should be taken into consideration when evaluating the immune response to any cellular product (Fig. 4).

The original source of the cell product has a potentially important impact. There are two common sources of pluripotent cells: hESCs and hiPSCs[84–87]. As hESCs and hiPSCs require different methodologies to generate the differentiated final cell product, it is, therefore, possible that they may express different antigens leading to different immunogenicity. Specifically, several factors present in the differentiation protocols and in vitro culture conditions may be sources of xeno or aberrant immunogenic antigens. Such expression can result from prolonged cultures in animal product-containing media, such as the uptake of the non-human sialic acid N-glycolyneuraminic acid (Neu5Gc) by hESC when cultured in standard conditions with serum[88–91]. Non-physiological media constituents, such as high concentrations of growth hormones or antibiotics have also been shown to induce the ectopic expression of CD30[92,93]. Finally, chromosomal abnormalities have been shown to be produced through in vitro propagation of hESC lines in multiple studies[94–97].

Another important confounder that may impact rejection is the expression of potentially immunogenic or immunomodulatory surface molecules and the release of soluble mediators. All these factors can be dependent on the differentiation or maturation state of the cells, independently of their cellular source. Specifically, accelerated in vitro differentiation processes might be associated with immunological immaturity, since it is

well established that somatic cells acquire such maturity during fetal development that continues after birth in a process mediated by intricate and gradual feedback mechanisms between the nascent immune system and developing somatic cells[98,99]. This process will dictate and shape the repertoire of activating and inhibitory signals that would then orchestrate the immune system. Such fine-tuning is unlikely to occur in vitro with methodologies that recapitulate long periods of normal development in just a few weeks, and also in the absence of biologically normal immune cell interactions. This may result in incompletely differentiated cells that may express ligands reminiscent of an embryonic origin that could potentially be retained and recognized as foreign antigens, such as early glycans, proteins, or other unique surface molecule modifications (e.g., TRA-1-81, TRA-1-60, SSEA3, SSEA4) that were not present during immune education of the recipient's immune system in an already-mature antigenic microenvironment[17–19]. As mentioned in previous sections, incompletely differentiated cells may also lack or exhibit a reduced expression of ligands that are essential for immune cell inhibition, including reduced expression of HLA-I (easily altered in in vitro conditions due to its complex regulation and response to inflammatory cytokines[20,21]), or absence/presence of specific KIR-ligands in the recipient's NK cells that were not properly educated to recognize certain HLA-I epitopes as self[22,100,101].

Conversely, expression of other ligands such as HLA-E and HLA-G, which are known to confer foeto-maternal tolerance, may be maintained and may protect the product against NK cell cytotoxicity[102]. Some of these mechanisms have been exploited by groups attempting to reduce cellular product rejection[27,28] (see "How can immunogenicity be overcome or ameliorated for the long-term success of cellular therapies?"). Other immunomodulatory mechanisms, reminiscent of those contributing to foeto-maternal tolerance could include: secretion of Arginase I resulting in decreased CD3 expression and lymphocyte proliferation[103], activation of hemoxygenase I enzyme to produce anti-inflammatory molecules[104], expression of high levels of cathepsin B and a serine protease inhibitor (serpin, SPI-9) that allows the destruction of granzyme-B released by T cells and NK cells[105], or expression of the Fas ligand (FasL, CD95L) resulting in T cell apoptosis[106–109].

In contrast to hESC-derived cells, the reprogramming procedure for the generation of hiPSCs could result in aberrant antigen presentation due to partial epigenetic memory retained from the

parental induced cells. This could produce chimeric surface molecules of more than one cell type, thus eliciting an immune response even in autologous transplants[66,110].

Some cellular therapies may require further differentiation or maturation in vivo through interaction with other immune or neighboring cells, as well as through exposure to blood flow or tissue mechanics. It is therefore likely that in some cases the immunogenicity captured in vitro may not fully recapitulate the immune response ultimately observed in vivo. Examples include stem cell-derived β-like cells needing in vivo cues from the islet niche to reach a fully mature and specialized state[111–113], or dopaminergic neurons that must undergo further maturation after transplantation in the midbrain compartment[114]. Of note, immunosuppression treatment may have a detrimental effect on this process, either directly or by modifying the immune micro-environment, but further work is required to quantify these effects.

Cellular therapies may be less prone to immune rejection where the derived cells inherently have an immunomodulatory immuno-protective function. Examples include cells acting as anatomical barriers (e.g., certain types of endothelial cells, astrocytes, or pericytes in the blood-brain barrier) or RPEs in the blood-retinal barrier (which endogenously express ligands such as PD-L1, CD95L, or secreted molecules as PEDF, TGF-β3, IL-10)[25,43–45]. Similarly, transplantation of cellular therapies in immune-privileged anatomical niches (e.g., eye, brain, testes, or placenta), may induce different immune responses. If not compromised by the specific disease or by the process of transplantation, such sites may confer an advantage for allogeneic donor cell engraftment.

### How can immunogenicity be overcome or ameliorated for the long-term success of cellular therapies?
A small number of early-stage hPSC-based clinical trials for multiple diseases are currently underway (Table 1). As classical immunosuppression is associated with a range of side effects[6], a number of strategies are being explored to reduce or eliminate the immunogenicity of stem cell-derived cellular therapies (Table 4).

**Immunosuppressive agents.** Immunosuppression is the current standard approach to prevent the rejection of solid organ transplants. This strategy has also been adopted by recent clinical trials to prevent the rejection of cellular therapies (Table 1). In general, there are five broad classes of immunosuppressant drugs: (i) glucocorticoids/steroids (e.g., prednisolone, dexamethasone,

**Table 4 Current strategies to ameliorate immune response upon transplantation of allogeneic cellular derivatives.**

| Strategy | Principle | Limitations | References |
|---|---|---|---|
| Immunosuppression  | Use of drugs targeting essential pathways for immune cell performance. Types: glucocorticoids/steroids, cytostatics, specific antibodies, drugs acting on immunophilins/other mechanisms | – Cytotoxic effects<br>– Vulnerable recipient's immune system | 115–119 |
| HLA matching  | HLA-typed cell banks that allow the matching of donor and recipient cells using homozygous cell lines with frequent HLA haplotypes. Examples: United Kingdom, Japan | – Minor alleles not matched<br>– Might require immunosuppression | 126–128 |
| Genetically modified Cells  | Genetically engineered cells with gene editing techniques (e.g., CRISPR/Cas9, TALENS) that would bypass specific immune mechanisms of action. Examples: HLA-I or/and -II knock out cell lines, HLA-C retained lines, immune cloaked cells | – Safety concerns:<br>● Off-target effects of targeting<br>● Conversion to malignant/infected cell that will not be recognized | 23,26,27,55,132 |
| Immune Tolerance  | A combination of deletion, cell-intrinsic checkpoints, and suppression by regulatory mechanisms. Examples: costimulatory and adhesion blockade, inhibitory ligand overexpression, adoptive cell therapy with naturally occurring Tregs/induced donor-specific Tregs | – Specificity of the inhibitory drug (costimulatory and adhesion blockade)<br>– Genetic engineering/viral off-targets (ligand overexpression)<br>– Cost and infrastructure needed per patient (adoptive therapy) | 47,143–147,150 |
| Cell Shielding  | Protection or shielding of the derived cells with specific materials or encapsulation devices. Examples: alginate-beads, functionalized hydrogels | – Immunocompatible materials<br>– Vascularization and function of the cells<br>– Permeability to essential soluble factors | 153,154,156–158,160,162,164–166<br><br>ViaCyte: https://viacyte.com/ |

hydrocortisone) that act by reducing inflammation; (ii) cytostatics (e.g., alkylating, antimetabolites, methotrexate, azathioprine, and mercaptopurine, cytotoxic antibiotics) that block T cell proliferation; (iii) specific antibodies (polyclonal, monoclonal, anti-T cell receptor, anti-IL-2 receptor) that deplete subsets of immune cells such as macrophages/APC, T cells and/or B cells; (iv) drugs acting on immunophilins (e.g., cyclosporine, tacrolimus, sirolimus, everolimus) that inhibit cytokine production by for instance calcineurin inhibitors; and (v) drugs with other mechanisms of actions (e.g., interferons, opioids, TNF binding proteins, mycophenolate, and small biological agents)[120,121].

While classical immunosuppressive drugs are an appealing choice to reduce rejection of cellular therapies, they are known to lead to potentially serious toxic side-effects, including increased risk of infection (e.g., viral, bacterial, or fungal), cancers (e.g., skin or lymphoma), and organ damage (e.g., renal failure)[6,122–124]. Moreover, classical immunosuppressive drugs do not specifically and effectively modulate the innate immune response. Also, it is unclear what impact, if any, immunosuppression may have on cell survival, maturation, and function following transplantation of the differentiated cells aimed for cellular therapy. Similarly, it is still not clear if a different regime would need to be used depending on the hESC or hiPSC origin of the cellular source. Until now, clinical studies using stem cell therapies either opted for local suppression (e.g., corticosteroids) or more general regimes of immunosuppression (e.g., cyclosporine, MMF, tacrolimus) (Table 4). These regimes have not been fine-tuned as they were aimed to maximally avoid donor cell rejection. Therefore, the use of immunosuppressive agents in the context of cellular therapies requires further investigation to bypass toxic side-effects in the recipients, and it is generally accepted that the development of more specific strategies to reduce immune-mediated rejection of cellular therapies is desirable.

**HLA-matching**. Clinical outcomes after solid-organ transplantation are dependent on the degree of HLA disparity between the donor and the recipient[7,125]. Development of HLA-typed cell banks that allow optimal matching of donor and recipient cells represents an attractive strategy to reduce the risk of immune rejection. This approach is particularly attractive for populations in which major HLA-alleles are relatively conserved. The required degree of matching will differ depending on the cell type of interest (e.g., whether HLA-II is expressed by the cells) and the site of transplantation. It is estimated that a total of 150 or 140 cell lines would be required to provide well-matched cellular therapies for 93% or 90% of the UK and Japanese populations, respectively[126–128]. It is important to note, however, that the generation of cell banks with sufficient diversity for large and heterogeneous populations such as the US will be challenging. Moreover, the risk of rejection would still remain due to the presence of mismatched minor HLA antigens. Also, patients with less common haplotypes would remain difficult to match and alternative strategies would be needed to prevent rejection. This approach has not been tested yet with regenerative cell therapies per se but the reduction in immune response using matched donors is evidenced in solid organ transplants[129].

**Genetically engineered cells**. The emergence of genetic engineering tools such as the CRISPR/Cas9 technology has enabled the pursuit of low- or non-immunogenic "universal" cellular therapies. This is predominantly explored through the elimination of HLA genes (individually or both HLA-I and HLA-II), most effectively by targeting essential molecules for their correct location and expression, namely β2M (required for proper surface expression of HLA-I) and CIITA or RFXANK (master

transcriptional regulators for HLA-II genes)[23,48,55]. Since HLA is the predominant driver of the alloimmune response, HLA-knock-out (HLA-KO) cells are expected to evade CD8⁺ and CD4⁺ T cell responses. However, HLA-I deficient cells may nonetheless activate recipient NK cells (triggered by the "missing-self" signal), thus posing an obstacle for successful transplantation. To overcome this, engineered cells overexpressing non-polymorphic HLA-I molecules such as HLA-E, HLA-F, and HLA-G, acting as inhibitors of NK cell-mediated lysis via NKG2A/CD94 (HLA-E), KIR2DL4, and ILT2 (HLA-G) receptors have been generated. This approach has been shown to have at least some efficacy in enabling allogeneic cells to escape immune attack in vitro and in xenotransplantation experiments, as shown with hPSC, hESC-RPE, and hESC-neural progenitor cells using NK cytotoxic/ $^{51}$Chromium release assays[27,28,130,131].

An alternative strategy is the generation of cell lines that express a common allele of a specific polymorphic HLA-I molecule (HLA-A, -B or -C) that should be matched and that would then be able to bind NK inhibitory receptors. For example, HLA-C retained cells that also lack HLA-II demonstrated the ability to suppress both NK and T cell responses while maintaining antigen presentation both in vitro and in vivo[55]. Another approach used in HLA-KO cells is the induction of overexpression of CD47 as a potent inhibitor of phagocytosis, thus avoiding macrophage and NK cell-mediated rejection in allogeneic hosts. This approach has been exemplified by demonstrating survival and proliferation of luciferase-labeled hiPSC and hiPSC-derivative allografts in immunocompetent mice[26]. Finally, "immune cloaking" is a more holistic approach that includes disruption of several pathways of immune activation and has been attempted by over-expression of immunomodulatory transgenes *PD-L1*, *FASL*, *CD47*, *CD200*, *CCL21*, *MFGE8*, *H2-M3*, and *SPI6* in mouse ESCs, which allowed them to survive indefinitely as allografts[132]. Universal non-immunogenic cells could also be engineered so as not to express auto- or neo-antigens that normally trigger responses in autoimmune or genetic diseases.

There are important outstanding safety and efficacy considerations that must be addressed in relation to genetically engineered cellular therapies. First, it is essential to demonstrate that the cells do not undergo malignant transformation or infection by virtue of their ability to evade immune recognition. The addition of a fail-safe/suicidal gene cassette system has thus been suggested to selectively eliminate cells that are, for example, more proliferative[133]. Second, it is important to demonstrate that the elimination of HLA (or other) molecules does not interfere with cell survival, differentiation, and function after transplantation. Despite the current high interest to adopt such technology by several pharmaceutical companies, the clinical translation of this promising approach is still immature and requires careful study in the context of a competent HIS to better anticipate the outcome of clinical trials.

**Induction of tolerance**. Immune tolerance is defined as the absence of a deleterious immune response despite the continued presence of the antigen (e.g., derived from the transplanted cells) in an intact immune system setting. It is a desirable state that could potentially be achieved through a combination of specific deletions, cell-intrinsic checkpoints, and suppression by regulatory mechanisms such as in vivo induction or expansion of Tregs[134–142]. The targeted manipulation of specific T cell co-stimulatory or inhibitory pathways by for instance inhibiting the second signal via specific monoclonal antibodies (e.g., anti-CD40L, CTLA4Ig, anti-LFA-1), might represent a way to induce tolerance[47,143]. Another approach to achieve tolerance that has

recently reached the clinic in solid organ transplantation[144–148] is ex vivo expansion of patient-derived polyclonal or donor-specific Tregs that would be transferred back to the recipient[149–152]. None of these approaches have been directly tested in the context of regenerative cell therapies. Nonetheless, they are promising strategies to efficiently regulate the recipient's immune system in the long term with minimal induction of toxic effects.

**Cell shielding/encapsulation.** Shielding of the cellular therapy with alginate-beads or specific encapsulation devices has been used for transplantation of β-like cells[153–156]. Excitingly, the company *ViaCyte* has developed different cell-encapsulated versions of SC-derived human pancreatic progenitor cells to treat Type 1 Diabetes, which has now entered Phase 2 clinical trials (https://viacyte.com/). However, the challenge remains on the maintenance of full oxygenation and vascularization of the cells for optimal function, while still providing sufficient immuno-protection over time. This has been largely achieved with materials of particular permeability and with a specific size and distribution of nanopores[157]. In addition, the use of such technology critically depends on the transplantation site and its accessibility to the recipient's immune system, which if protected (e.g., immunoprivileged sites), might confer natural tolerance of the grafted cells. Finally, a potentially more refined approach would be to incorporate biocompatible materials in the matrix that contains molecules (e.g., anti-inflammatory cytokines, blocking antibodies) that would protect the cell product against the recipient's immune response. This technology is still in its infancy, but in proof-of-concept studies, hydrogels functionalized with IFN-γ, TGF-β1, or IL-10 siRNA were able to effectively modulate the immune response to several immune cells types[158–166]. This represents an elegant strategy free of both in vitro and ex vivo manipulation, although the challenge might be keeping the cellular therapy free from immune invasion as the biocompatible material degrades.

In conclusion, while none of the above-mentioned approaches are perfect on their own, they may be combined synergistically to minimize immunogenicity. An example of such a combined strategy may be to use engineered cells or HLA-typed cells transplanted with low-dose immunosuppression or with strategies to induce tolerance while under a specific shielding material. Such combinations would be patient-dependent (e.g., if suffering from a precondition that prevents the use of immunosuppressants), cellular therapy-dependent (e.g., if a cell type is specially affected by the genetic manipulation or the immunosuppressive regime), and site-dependent (e.g., in an immune-privileged site, use of shielding material may not be necessary). In summary, a bespoke and multi-dimensional case-specific approach is likely to be necessary to avoid graft rejection while guaranteeing the survival and function of the specific cellular therapy.

## Outlook

**Concluding remarks: the need for multi-disciplinary approach and infrastructure for immune-assessment of cellular therapies.** As outlined in this review, characterizing and modulating the immune response to regenerative cellular therapies is an essential step for their clinical translation. The immune response is highly complex, consisting of multiple overlapping and interconnected pathways (Table 2). A combination of in vitro and in vivo methodologies must, therefore, be used in concert for the definitive evaluation of both the innate and adaptive immune responses (Table 3). Importantly, all experimental paradigms have inherent weaknesses and advantages, and extrapolation of the findings to the clinical scenario should be made with caution. We described several confounders that can affect the conclusions

of immunogenicity studies, including the original cell source, differentiation procedure, the identity of the cell type of interest, maturation status, site of transplantation, or methodological limitations (Fig. 4). This complexity is mirrored by the plethora of approaches under investigation to ameliorate or eliminate the immune response to cellular therapies (Table 4). In view of the redundancy displayed by the immune system, it is likely that a combined methodology is would be the most feasible and cost-effective treatment modality.

We conclude that the immunological assessment of cellular therapies is as necessary as functional or safety verification prior to clinical translation. Few research groups possess the complete range of resources, experimental models, and expertise to conduct the full breadth of studies required to generate definitive and validated immunogenicity data that would be relevant to multiple cellular therapies. We, therefore, suggest that a reasonable and practical approach would be the establishment of collaborative networks and platforms combining diverse expertise and infrastructure, which would warrant standardized methods for their immunogenicity assessment (in a similar way to the recent global initiative *ARDAT* for advanced cell and gene therapies: http://ardat.org/). If supported by the full breadth of in vitro and in vivo experimental models currently available, such a consortium approach would be well placed to plan and execute definitive immunogenicity studies that ultimately ensure the safe and efficient clinical translation of any cellular product.

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

## Acknowledgments

This work was supported by the UK Regenerative Medicine Platform (UKRMP, [Grant number MR/S020934/1]). This research was also funded in part, by the Wellcome Trust [Grant number RG79413]. For the purpose of open access, the author has applied a CC BY public copyright licence to any Author Accepted Manuscript version arising from this submission. Figure drawings were created with BioRender.com.

## Author contributions

S.P.-R. and M.R. wrote the manuscript. K.S.P., G.L., J.L.J., and S.H. revised and edited the text. S.P.-R. made the figures. All authors contributed to discussions and structure definitions of the respective sections.

## Competing interests

The authors declare no competing interests.
