## [Peer Review File · Communications Biology]

Reviewers' comments:

Reviewer #1 (Remarks to the Author):

Summary and general comments:

In the review article "Immunological considerations and challenges for regenerative cellular therapies" Petrus-Reurer et al. describe the topic of regenerative medicine using stem cell therapy along with the potential immunogenicity of stem cells as one of the challenges. The review mainly focuses on three areas including: (1) Innate and adaptive immune responses and their role in graft rejection, (2) In vitro and in vivo approaches to examine allergenicity in general, (3) potential strategies to ameliorate immune rejection. The authors covered these topics elegantly; however, the review lacks deep discussion and speculation on stem cell therapy in these areas. The authors nicely pose a question before each paragraph; however, it will be better if the authors rephrase the question towards the review topic (please see the example below-point#3). I really enjoyed reading the tables summarizing the clinical trials, methodologies, and current strategies to ameliorate allo-immune responses. Overall, the review contains wealth of information discussing the above-mentioned areas; however, it requires more speculation and deep discussion from the literature about stem cell therapy and its connection to the phenomenon of allo-graft rejection.

Specific comments:

Abstract:

Line 26-27 "we also discuss how the immune response is determined...." This is very interesting area of investigation albeit has not been discussed/highlighted clearly in the review.

Main text:

1. Introduction: It will be better if the authors start the review with a short introduction to introduce non-expert readers to the field of stem cell therapy before posing the first question. The following points are suggestion for brief introduction:

- Definition of regenerative medicine.
- Type of Stem cells
- Source of stem cells (Iso vs Allografts).
- Application of stem cell therapy in different diseases (e.g., AMD, Parkinson disease..etc.)
- Promises (immune-privilege organs) and Challenges (which include rejection). This final point will be a nice sag-way for the immunogenicity as a potential challenge.

2. First question: "Why is understanding.....?"

The authors nicely pose the question at the beginning of section albeit the paragraph did not answer it clearly. It will be nice if the authors rephrase it by adding "immunogenicity to cell therapy". Lines 42-48 and line 314-315 could be used for the brief introduction. Answering the question could also include incidence and prevalence of failed trials/procedures of stem cell therapy (if it is available). Hence, understanding immunogenicity to stem cell therapy will be an important topic to study albeit it is challenging to rule out immunogenicity as a contributing factor because of other factors that could contribute to failure of the procedure. Potency should be used instead of potentiality (line 45).

3. Second question: "What are the dominant drivers.....?"

The question could be rephrased towards the review topic. For instance, "Does innate immune response play a role in rejection of stem cell therapy? Or something similar. The authors answer the question very nicely by giving a general introduction about the host innate and adaptive immune responses as triggers of allograft rejection. However, the authors did not describe or speculate deeply the potential interaction of stem cells with both arms of the immune system. For instance, lines 71-87 describe the types of innate cells and their defense mechanisms while only lines 88-89 and 102-103 very briefly speculate on the role of these cells at the time of infusion of stem cells. It will be nice if the authors briefly describe the general role of innate immune cells

then speculate/discuss more about their role in stem cell rejection. For example, they can discuss the expression of MHC-I by stem cells e.g., hPSC and the possibility that NK cells will attack them because of low expression of MHC-I by these cells.

Regarding the adaptive immune response, the authors did not discuss the term "allo-reactive T cells" and their origins. Further, the authors should discuss/cite from the literature possible consequences regarding interaction between T cells [CD4 vs CD8] and hPSC. The conclusions from lines 173-176 should be interpreted carefully and consider re-writing because embryonic antigens (TRA-1, SSEA3...etc) expressed by hPSC even in isograft setting as mentioned under question 4 might elicit T cell responses.

4. Third question: What are the available experimental platforms to study immunogenicity....? Again, the authors covered very nicely a wide-spectrum of methodologies but deep discussion is missing regarding stem cell therapy. For instance, lines 206-208 need more explanation and speculation.

5. Fourth question: What are the immunological confounders...? The authors nicely combine the possible events that can occur during differentiation of stem cells both in vitro or in vivo. However, none of these events have been connected or discussed in the context of immunogenicity and rejection of stem cells. Different paragraphs under this question could be reshuffled and used in other sections. For instance, does any of the embryonic antigens (TRA-1, SSEA3...etc) expressed by hPSC can elicit immune response via activation of T cells? This could be a good discussion point under the second question regarding T cells and allo-graft rejection. Also, lines 344-348 could be another discussion/speculation point under the role NK cells in the same process. In conclusion, there is wealth of information in this section but requires redistribution across the different section in the right context.

6. Fifth question: How can immunogenicity be overcome.....? It will be nice if the authors discuss the effect of immunosuppressive drugs on in vivo differentiation of hPSC to the target cell of requirement. Also, is there is a differential requirement for immunosuppression regimens when we use hESC vs hPSC? Further, can we achieve tolerance of the cell therapy by injection into immune privilege sites? Finally, is there is a possibility to use combination of these approaches, is there is a benefit for one over another, or certain approaches will be suitable for patient compared to another. For instance, a patient with Age-related macular degeneration (AMD) could be a good candidate for tolerance induction approach since the stem cells will be injected via Anterior chamber of the eye.

Tables:

1. In general, it will be nice to add references to the table. Couple of references were added in table 1 (Mousset et al.).
2. Adding a separate column to the table regarding the kind of disease or injury will be nice. Also, if any of these clinical trials were either published or terminated (reason behind termination).
3. Typos in table 1: HLE-E.

Figures: (Note: Figures numbering has not been written in the figure itself).

1. Figure 1: It will be nice if the authors include in the figure stem cells and their potential interaction with NK cells and lymphocytes.

Reviewer #2 (Remarks to the Author):

Summary:

In their review article entitled, "Immunological considerations and challenges for regenerative cellular therapies", the authors attempt to shed light on the complex issue of immunogenicity of regenerative cellular therapies and how to overcome it. They set out to discuss four main topics: basic immunology of rejection, assay platforms to investigate sources of immunogenicity, confounding factors that may influence the degree of immunogenicity, and lastly, strategies that can be used to overcome immunogenicity.

Overall assessment: While certainly a timely subject, the review article falls short of providing the Nature Communications reader with sufficient insight. It only superficially mentions key studies to link the hESC/hiPSC cell therapy field and the topic of immunogenicity. Often times, this is accomplished with a generic sentence at the end of a longer paragraph. The authors should take the time to describe key findings with cell therapies in the context of each assay or concept. It may be helpful to compare and contrast findings where different strategies to overcome immunogenicity have been employed, rather than glossing over the different approaches in a few short sentences. A major revision is recommended in order to provide more insightful descriptions of immunogenicity considerations and challenges for regenerative cell therapies. Specific comments are listed below.

Specific comments:

1. The first third of the article reads like a basic immunology textbook with little if any real connection to regenerative cellular therapies, save a single sentence which alludes to hiPSC-RPE. The authors may be better served by referencing a text book or other review article which covers basic tenants of innate and adaptive immunity in greater detail. Then, they could spend more time relating such tenants to experimental findings for regenerative cell therapies.
2. The in vitro methodologies section reads like a laundry list of assays with little mention of how cellular therapies have performed in such assays other than just to say "...some studies have assessed the expression of activating or inhibitory ligands" or "co-cultures can be executed". The laundry list of assays could be covered by Table 1, while the manuscript text could be better used to discuss specific examples of how cellular products (eg, by focusing on one or two specific cell types in clinical trials) have performed in some of the most commonly used assays.
3. The in vivo section again cites a laundry list of endpoints that can be used. Why not describe the results of a few key studies where regenerative cell therapies have been used in such models and what endpoints were used to assess immunogenicity?
4. [Minor] Line 305... "or behavioral assays for dopaminergic cells"... this phrase likely should have been included within the parentheses yet falls outside.
5. Confounders section... this is better than the preceding sections, as it does reference various hESC/hiPSC specific studies. Can some of these be discussed in greater detail to exemplify a point here or there?
6. Immune evasion strategies section, please describe in more detail how each approach has been applied to regenerative cell therapies and what the outcome was. This section allows you to draw upon concepts covered in the preceding sections (immune effector cells, assays, in vivo models) and tying them to the strategies for evading immune detection.
7. Concluding remarks.... Add in a statement about how standardization of methods for assessing immunogenicity of cell therapies may be warranted. For example, a recent global initiative called ARDAT attempts to do so for cell and gene therapies (<https://www.imi.europa.eu/projects-results/project-factsheets/ardat>) and may be timely to mention in this section.
8. Comments on Figures:
Figure 2: where is the target cell? Can you relate the allorecognition to the target cell (cellular therapies) graphically?
Figure 3: lovely figure
Table 1: great table and an appropriate place for a laundry list of assays, can you list examples of specific references where such assays have been used with regenerative cell therapies?
Table 2: seems out of place and the column listing the "format" is not necessary since none of these are discussed in any great detail in the article
Table 3: a nice table, can you add examples of specific references where such strategies have been employed either clinically or preclinically with each strategy?

RESPONSE TO REVIEWERS

Immunological considerations and challenges for regenerative cellular therapies

S. Petrus-Reurer, M. Romano, S. Howlett, J.L. Jones, G. Lombardi and K. Saeb-Parsy

Referee expertise:

Referee #1:

Referee #2:

Referee #3:

Reviewers' comments:

Reviewer #1 (Remarks to the Author):

Summary and general comments:

In the review article "Immunological considerations and challenges for regenerative cellular therapies" Petrus-Reurer et al. describe the topic of regenerative medicine using stem cell therapy along with the potential immunogenicity of stem cells as one of the challenges. The review mainly focuses on three areas including: (1) Innate and adaptive immune responses and their role in graft rejection, (2) *In vitro* and *in vivo* approaches to examine allergenicity in general, (3) potential strategies to ameliorate immune rejection. The authors covered these topics elegantly; however, the review lack deep discussion and speculation on stem cell therapy in these areas. The authors nicely pose a question before each paragraph; however, it will be better if the authors rephrase the question towards the review topic (please see the example below-point#3). I really enjoyed reading the tables summarizing the clinical trials, methodologies, and current strategies to ameliorate allo-immune responses. Overall, the review contains wealth of information discussing the above-mentioned areas; however, it requires more speculation and deep discussion from the literature about stem cell therapy and its connection to the phenomenon of allo-graft rejection.

We thank Reviewer #1 for the constructive comments. We indeed appreciate that the review was lacking more speculation and discussion regarding stem cell therapies in the different sections presented, which now has been incorporated and, in our opinion, has increased the focus of the review manuscript on immunogenicity of stem cell therapies.

Specific comments:

Abstract:

Line 26-27 "we also discuss how the immune response is determined....." This is very interesting area of investigation albeit has not been discussed/highlighted clearly in the review.

We appreciate that indeed this specific point was not discussed in enough depth, so we have now reformatted section II (see in point 3 the description of the specifics) to address this point more clearly.

Main text:

1. Introduction: It will be better if the authors start the review with a short introduction to introduce non-expert readers to the field of stem cell therapy before posing the first question. The following points are suggestion for brief introduction:

- Definition of regenerative medicine.
- Type of Stem cells
- Source of stem cells (Iso vs Allografts).
- Application of stem cell therapy in different diseases (e.g., AMD, Parkinson disease..etc.)
- Promises (immune-privilege organs) and Challenges (which include rejection). This final point will be a nice sag-way for the immunogenicity as a potential challenge.

We thank the reviewer for the suggestion. We have now revised section I and incorporated these points, while also addressing point 2 of the reviewer.

2. First question: "Why is understanding.....?"

The authors nicely pose the question at the beginning of section albeit the paragraph did not answer it clearly. It

will be nice if the authors rephrase it by adding “immunogenicity to cell therapy”. Lines 42-48 and line 314-315 could be used for the brief introduction. Answering the question could also include incidence and prevalence of failed trials/procedures of stem cell therapy (if it is available). Hence, understanding immunogenicity to stem cell therapy will be an important topic to study albeit it is challenging to rule out immunogenicity as a contributing factor because of other factors that could contribute to failure of the procedure. Potency should be used instead of potentiality (line 45).

We thank the reviewer for the constructive comment. We have now modified the title for: “I. Introduction: why is understanding immunogenicity of a cell therapy important?” We added a paragraph addressing the current state of stem cell therapies trials (lines 49-53), which indeed helps introducing the immunogenicity of the therapies as main topic of the review and current biggest challenge that stem cell therapies are facing to successfully reach the clinics. “Potentiality” has now been changed for “potency” (line 45).

3. Second question: “What are the dominant drivers.....?”

The question could be rephrased towards the review topic. For instance, “Does innate immune response play a role in rejection of stem cell therapy? Or something similar. The authors answer the question very nicely by giving a general introduction about the host innate and adaptive immune responses as triggers of allograft rejection. However, the authors did not describe or speculate deeply the potential interaction of stem cells with both arms of the immune system. For instance, lines 71-87 describe the types of innate cells and their defense mechanisms while only lines 88-89 and 102-103 very briefly speculate on the role of these cells at the time of infusion of stem cells. It will be nice if the authors briefly describe the general role of innate immune cells then speculate/discuss more about their role in stem cell rejection. For example, they can discuss the expression of MHC-I by stem cells e.g., hPSC and the possibility that NK cells will attack them because of low expression of MHC-I by these cells.

Regarding the adaptive immune response, the authors did not discuss the term “allo-reactive T cells” and their origins. Further, the authors should discuss/cite from the literature possible consequences regarding interaction between T cells [CD4 vs CD8] and hPSC. The conclusions from lines 173-176 should be interpreted carefully and consider re-writing because embryonic antigens (TRA-1, SSEA3...etc) expressed by hPSC even in isograft setting as mentioned under question 4 might elicit T cell responses.

We thank the reviewer for this very valuable suggestion on how to re-structure the section to present a more speculative read on drivers triggering immune response to stem cell therapies.

We have first changed the title for: “II. What are the dominant drivers of immune response to stem cell therapies?”. We have then re-written the section before describing general mechanisms of immune-target cell interactions (at present summarized in new Table 1), now more focusing on specific examples and speculation on how cellular therapies can elicit both innate and adaptive responses that would lead to graft rejection. We also described the term and origins of “allo-reactive T cells”, discussed the possible consequences of T cell and hPSC interaction, and speculated on immunosuppressive mechanisms relevant to cellular therapies (see new section II). We have also re-written lines 173-176 to improve clarity (lines 126-135).

4. Third question: What are the available experimental platforms to study immunogenicity....? Again, the authors covered very nicely a wide-spectrum of methodologies but deep discussion is missing regarding stem cell therapy. For instance, lines 206-208 need more explanation and speculation.

We thank the reviewer for this suggestion. We have now added more explanation and discussion describing specific studies for each specific immune compartment *in vitro* and *in vivo* and with additional references to stem cell studies that used the described methodologies (also added to Table 2).

5. Fourth question: What are the immunological confounders...? The authors nicely combine the possible events that can occur during differentiation of stem cells both *in vitro* or *in vivo*. However, none of these events have been connected or discussed in the context of immunogenicity and rejection of stem cells. Different paragraphs under this question could be reshuffled and used in other sections. For instance, does any of the embryonic antigens (TRA-1, SSEA3...etc) expressed by hPSC can elicit immune response via activation of T cells? This could be a good discussion point under the second question regarding T cells and allo-graft rejection. Also, lines 344-348 could be another discussion/speculation point under the role NK cells in the same process. In conclusion, there is wealth of information in this section but requires redistribution across the different section in the right context.

We thank the reviewer for bringing up this point and we appreciate that some parts of the section could benefit from being mentioned in other sections. We have therefore now added some elements (e.g. reminiscent embryonic antigens in cellular therapies or lack/mismatch of HLA-I that would trigger T cell and NK cells, respectively) in section II, which nicely sets the ground for a more founded discussion on the immunological confounders that should be taken into consideration for stem cell therapies (section IV). As this section was praised by one of the other reviewers, we have kept the section and tried to incorporate suggestions from both reviewers.

6. Fifth question: How can immunogenicity be overcome.....? It will be nice if the authors discuss the effect of immunosuppressive drugs on *in vivo* differentiation of hPSC to the target cell of requirement. Also, is there is a differential requirement for immunosuppression regimens when we use hESC vs hPSC? Further, can we achieve tolerance of the cell therapy by injection into immune privilege sites? Finally, is there is a possibility to use combination of these approaches, is there is a benefit for one over another, or certain approaches will be suitable for patient compared to another. For instance, a patient with Age-related macular degeneration (AMD) could be a good candidate for tolerance induction approach since the stem cells will be injected via Anterior chamber of the eye.

We thank the reviewer for mentioning these important points which strengthen section V. We have now further discussed (i) the still unknown effect of immunosuppressive drugs on *in vivo* differentiation of hPSC to target cell, and (ii) if a differential requirement for immunosuppression would be needed for different stem cell sources (lines 443-448). We now have also mentioned the possibility of increased tolerability of the cell therapy dependent on site of injection (lines 538-539). In addition, we described in more detail how each approach to overcome/ameliorate immunogenicity has been applied to regenerative cell therapies and its outcome, and we also emphasized the strengths and difficulties of each strategy. Finally, we added a final paragraph mentioning the benefit of a combinatorial approach with specific examples as such combination would most likely be patient-, cellular therapy- and site-dependent.

Tables:

1. In general, it will be nice to add references to the table. Couple of references were added in table 1 (Mousset et al.).

We agree with the reviewer. Specific references have now been added to new Table 2.

2. Adding a separate column to the table regarding the kind of disease or injury will be nice. Also, if any of these clinical trials were either published or terminated (reason behind termination).

We agree with the reviewer. Two new extra columns have been added to new Table 3 stating the specific "Disease" and "Status" of the trial. We also added specific references in case there were published studies for any of the described trials.

3. Typos in table 1: HLE-E.

Corrected in new Table 2.

Figures: (Note: Figures numbering has not been written in the figure itself).

Corrected in all figures.

1. Figure 1: It will be nice if the authors include in the figure stem cells and their potential interaction with NK cells and lymphocytes.

For clarification, we have now renamed "Target Cell" for "Cellular Therapy (Target Cell)" in Figure 1. Interactions of the cellular therapy with lymphocytes (T cells) are explicitly represented in Figure 2, which has now been re-formatted for clarity.

Reviewer #2 (Remarks to the Author):

Summary:

In their review article entitled, “Immunological considerations and challenges for regenerative cellular therapies”, the authors attempt to shed light on the complex issue of immunogenicity of regenerative cellular therapies and how to overcome it. They set out to discuss four main topics: basic immunology of rejection, assay platforms to investigate sources of immunogenicity, confounding factors that may influence the degree of immunogenicity, and lastly, strategies that can be used to overcome immunogenicity.

Overall assessment: While certainly a timely subject, the review article falls short of providing the Nature Communications reader with sufficient insight. It only superficially mentions key studies to link the hESC/hiPSC cell therapy field and the topic of immunogenicity. Often times, this is accomplished with a generic sentence at the end of a longer paragraph. The authors should take the time to describe key findings with cell therapies in the context of each assay or concept. It may be helpful to compare and contrast findings where different strategies to overcome immunogenicity have been employed, rather than glossing over the different approaches in a few short sentences. A major revision is recommended in order to provide more insightful descriptions of immunogenicity considerations and challenges for regenerative cell therapies. Specific comments are listed below.

We thank the Reviewer #2 for the constructive feedback. We indeed appreciate that the review was falling short in providing more speculation and discussion regarding immunogenicity of stem cell therapies throughout the different described sections. We have now revised the manuscript according to the suggestions and have incorporated key findings, examples and a more speculative narrative on the subject, increasing the focus of the review on immunogenicity of stem cell therapies.

Specific comments:

1. The first third of the article reads like a basic immunology textbook with little if any real connection to regenerative cellular therapies, save a single sentence which alludes to hiPSC-RPE. The authors may be better served by referencing a text book or other review article which covers basic tenants of innate and adaptive immunity in greater detail. Then, they could spend more time relating such tenants to experimental findings for regenerative cell therapies.

We thank the reviewer for this comment. We agree that section II was rather descriptive. We have now moved the descriptive content to new Table 1, and the section has been re-written focusing on specific examples and speculation on how cellular therapies can elicit both innate and adaptive responses that would lead to graft rejection. We have also discussed the possible mechanisms and consequences of immune and hPSC cell interaction, and speculated on immunosuppressive mechanisms present in some cellular therapies related to experimental findings for regenerative cellular therapies.

2. The *in vitro* methodologies section reads like a laundry list of assays with little mention of how cellular therapies have performed in such assays other than just to say ... “some studies have assessed the expression of activating or inhibitory ligands” or “co-cultures can be executed”. The laundry list of assays could be covered by Table 1, while the manuscript text could be better used to discuss specific examples of how cellular products (eg, by focusing on one or two specific cell types in clinical trials) have performed in some of the most commonly used assays.

We thank the reviewer for the constructive comment. In this section, in addition to describe the most commonly used assays to evaluate each specific immune cell compartment *in vitro*, we have now illustrated the text with specific examples and results from pivotal studies that assess the immunogenicity of their generated PSC-cellular products. In addition, we have expanded the references for each immune compartment in the text, and also individually referenced the *in vitro* methods described in new Table 2.

3. The *in vivo* section again cites a laundry list of endpoints that can be used. Why not describe the results of a few key studies where regenerative cell therapies have been used in such models and what endpoints were used to assess immunogenicity?

We thank the reviewer for the suggestion. In the revised manuscript, we attempted to balance a more comprehensive survey of endpoints with specific examples of studies that exemplify methods of assessments. We have now clearly referenced each specific endpoint for further consultation, and also cited and exemplified the paragraph regarding complementary methods to analyze immune infiltration, graft cell survival and/or function of the cellular therapy in the site of injection (lines 332-334). Specific references have also been added for the *in vivo* methods described in new Table 2.

4. [Minor] Line 305... “or behavioral assays for dopaminergic cells”... this phrase likely should have been included within the parentheses yet falls outside.

We thank the reviewer for the minor comment. In fact, it was meant to be outside of the parenthesis as we refer to real-time imaging techniques or behavioral assays as complementary assays to assess immune infiltration, graft cell survival and/or function of the cellular therapy in the site of injection. Examples for real-time imaging techniques would then include the use of luciferase constructs or SD-OCT scans. We now edited the sentence for clarity (lines 328-332).

5. Confounders section... this is better than the preceding sections, as it does reference various hESC/hiPSC specific studies. Can some of these be discussed in greater detail to exemplify a point here or there?

We thank the reviewer for the suggestion. We have now added some more details in specific parts of the text of section IV.

6. Immune evasion strategies section, please describe in more detail how each approach has been applied to regenerative cell therapies and what the outcome was. This section allows you to draw upon concepts covered in the preceding sections (immune effector cells, assays, *in vivo* models) and tying them to the strategies for evading immune detection.

We thank the reviewer for the comment. We have now included for each strategy, how it has been applied to regenerative cell therapies and with its outcome if applicable. We have also emphasized the strengths and difficulties of each approach while tying them to concepts covered in the preceding sections. Finally, we have added a final paragraph mentioning the benefit of a combinatorial approach with specific examples as such combination would most likely be patient-, cellular therapy- and site-dependent.

7. Concluding remarks.... Add in a statement about how standardization of methods for assessing immunogenicity of cell therapies may be warranted. For example, a recent global initiative called ARDAT attempts to do so for cell and gene therapies (<https://www.imi.europa.eu/projects-results/project-factsheets/ardat>) and may be timely to mention in this section.

We thank the reviewer for this timely suggestion. We have now mentioned a statement in section VI about how the establishment of collaborative networks and platforms combining diverse expertise and infrastructure would warrant standardized methods for their immunogenicity assessment, including the reference to the global initiative ARDAT for advanced cell and gene therapies (lines 585-587).

8. Comments on Figures:

Figure 2: where is the target cell? Can you relate the allorecognition to the target cell (cellular therapies) graphically?

Thank you for the comment on Figure 2. We have now re-drawn the figure for clarity of the pathways of allorecognition, and we have expanded the figure legend also indicating that “Donor DC/Cellular Therapy” refers to an “HLA-II expressing target cell”. In addition, we have illustrated the two processes of Cellular and Antibody-mediated rejection of the cellular therapy.

Figure 3: lovely figure

Thank you.

Table 1: great table and an appropriate place for a laundry list of assays, can you list examples of specific references where such assays have been used with regenerative cell therapies?

We agree with the reviewer. References have now been added to new Table 2.

Table 2: seems out of place and the column listing the “format” is not necessary since none of these are discussed in any great detail in the article

We thank the reviewer for the suggestion, but we decided to keep it in as although is not directly discussed in the text it might be valuable information for the reader.

Table 3: a nice table, can you add examples of specific references where such strategies have been employed either clinically or preclinically with each strategy?

We agree with the reviewer. Specific references have now been added to new Table 4.

REVIEWERS' COMMENTS:

Reviewer #1 (Remarks to the Author):

It is very clear from the revised version that the authors put the effort and thinking to enhance the review. They addressed all the mentioned comments. One minor editing issue is related to abbreviations. For instance, the full definition of RPE in line 122 as well as PRRs in line 198 has not been mentioned. I think the review will be a nice addition at the interface of regenerative cell therapy and transplantation immunology.

Reviewer #2 (Remarks to the Author):

The revised manuscript is substantially improved over the original version. Section II is far better, with descriptive content being displayed through new Table 1. The authors have, in many (but not all) sections, also done a good job of highlighted key findings from published studies to exemplify main points. Authors have added references to the figures, which is also helpful. Certain sections could still stand to be improved/focused a bit more, as the language remains very generic and descriptive. It is well appreciated how much work has gone into the revisions made to date, yet a few more line-item revisions, such as those listed below, would substantially improve the manuscript and warrant publication in Nature Communications. If some of these items can be addressed, the manuscript could be recommended for acceptance.

Specific points:

1. Introduction, line 52-53, the text cites "more than 20 stem cell based studies are registered on ClinicalTrials.gov but only a few have published results hence outcomes are awaited". Please cite the "published results" that are mentioned in this sentence as it would be helpful to the reader to know which studies are being called out here. Also, as the focus is on immunogenicity, what sort of immunosuppression has been used in these studies so far? Can the authors summarize what these initial results have shown thus far (ie, in general, only safety, no efficacy data yet?, any evidence of immunogenicity yet?)...Also, on a side note, this would be an appropriate place to cite the Table which lists these trials (ie, it is now Table 3).
2. Line 141-143, can the authors elaborate a little more on this key example? Were the PD-L1 overexpressing cells compared to non-engineered cells? Simply adding an extra phrase to the end of the sentence would be enough. How much better did the PD-L1 expressing cells perform than the non-engineered version?
3. Line 145-147, the authors list 6 references but give no specific examples or details to the reader to help provide insight. Some additional information would go a long way. Eg, what type of inflammatory molecules were used, what are the various cell types that have shown class II upregulation in these studies? Are there any examples of lineages where class II upregulation was not noted?
4. Line 147-8, the authors state that several groups have knocked out HLA genes yet only 2 references (#23 and 48) are given. Why these two? They are not the first studies to knock out HLA genes, so it may be helpful to state that since X year, several studies have genetically engineered cells to knock out HLA genes. A historical perspective may help the reader understand how long the field has been studying the impact of knocking out HLA genes on immunogenicity.
5. Line 189, what were the "respective gene editing approaches" that were used in the 4 studies? Any differences in how well the approaches worked? Can the authors speculate as to the relative merits of these different approaches?
6. Line 193-5... what derivative cell types have been studied in the 4 references provided? Which of the 12 activating or inhibitory ligands were evaluated on which cell type? A little more color would be helpful here. eg, "For example, Smith et al assessed XX and found YY. Jones et al did ZZZ.

7. Line 198-210. This paragraph provides three specific examples and gives good comparative context. It shows the type of context that would be helpful in other sections. One small note, "PPRs written instead of PRRs"

8. Line 259, what are the "discrepancies" between the two studies? Can you be more specific?

9. Line 306-326. This section needs work. It is very dense. Can you break it up with a few specific examples for at least a couple of the assays being mentioned? What were the results? Which cell types have been studied and using which assays? What were key findings? Did any studies compare in vivo immunogenicity of two different cell types?

10. Line 485-6, what assays were used and what lineages were studied in the 4 referenced papers?

11. Line 488-498- what assays were used in some of the references cited and what were the results?

12. Table 4, "Genetically Modified Cells" row, please ensure all the appropriate references that are mentioned in the text are also cited here (eg, missing ref 132 for immune cloaking?, please check)

REVIEWERS' COMMENTS:

Reviewer #1 (Remarks to the Author):

It is very clear from the revised version that the authors put the effort and thinking to enhance the review. They addressed all the mentioned comments. One minor editing issue is related to abbreviations. For instance, the full definition of RPE in line 122 as well as PRRs in line 198 has not been mentioned. I think the review will be a nice addition at the interface of regenerative cell therapy and transplantation immunology.

We thank the reviewer for all the useful comments that truly helped improving the content and structure of the present review article manuscript.

Reviewer #2 (Remarks to the Author):

The revised manuscript is substantially improved over the original version. Section II is far better, with descriptive content being displayed through new Table 1. The authors have, in many (but not all) sections, also done a good job of highlighted key findings from published studies to exemplify main points. Authors have added references to the figures, which is also helpful. Certain sections could still stand to be improved/focused a bit more, as the language remains very generic and descriptive. It is well appreciated how much work has gone into the revisions made to date, yet a few more line-item revisions, such as those listed below, would substantially improve the manuscript and warrant publication in Nature Communications. If some of these items can be addressed, the manuscript could be recommended for acceptance.

We thank the reviewer for all the constructive comments that truly helped shaping and improving the review article manuscript. We also thank the reviewer for highlighting the text, table and figure improvements included in the previous revision. We have now addressed all remaining raised minor comments as specified in the point-by-point response below, which we hope will fully satisfy the reviewer.

Specific points:

1. Introduction, line 52-53, the text cites “more than 20 stem cell based studies are registered on [ClinicalTrials.gov](https://clinicaltrials.gov) but only a few have published results hence outcomes are awaited”. Please cite the “published results” that are mentioned in this sentence as it would be helpful to the reader to know which studies are being called out here. Also, as the focus is on immunogenicity, what sort of immunosuppression has been used in these studies so far? Can the authors summarize what these initial results have shown thus far (ie, in general, only safety, no efficacy data yet?, any evidence of immunogenicity yet?)...Also, on a side note, this would be an appropriate place to cite the Table which lists these trials (ie, it is now Table 3).

We thank the reviewer for this comment. We have now mentioned the initial results of these studies in lines 52-54, and moved Table 3 to this paragraph (now new Table 1). This table includes the references to the studies that have published results, and the type of immunosuppression used in each trial if the information was available. The rest of the tables have been re-referenced accordingly.

2. Line 141-143, can the authors elaborate a little more on this key example? Were the PD-L1 overexpressing cells compared to non-engineered cells? Simply adding an extra phrase to the end of the sentence would be enough. How much better did the PD-L1 expressing cells perform than the non-engineered version?

We have now elaborated on this example/reference 47 in lines 146-149.

3. Line 145-147, the authors list 6 references but give no specific examples or details to the reader to help provide insight. Some additional information would go a long way. Eg, what type of inflammatory molecules were used, what are the various cell types that have shown class II upregulation in these studies? Are there any examples of lineages where class II upregulation was not noted?

We have now provided the requested details in line 151. We have not elaborated further as the cell type studied in the referenced studies were hPSC-RPE cells, which showed HLA-II upregulation upon the presence of inflammatory molecules.

4. Line 147-8, the authors state that several groups have knocked out HLA genes yet only 2 references (#23 and 48) are given. Why these two? They are not the first studies to knock out HLA genes, so it may be helpful to state that since X year, several studies have genetically engineered cells to knock out HLA genes. A historical perspective may help the reader understand how long the field has been studying the impact of knocking out HLA genes on immunogenicity.

We thank the reviewer for the comment. We have now indicated in the text the historical interest of the field to study the impact of knocking out HLA genes on immunogenicity (lines 154-158). The referenced publications were selected based on pivotal studies that emerged in the recent times specifically aimed to generate cells for replacement therapies, and in light of the recent advances in genetic engineering tools that facilitate the editing of HLA genes. This has now also been specified in the respective paragraph.

5. Line 189, what were the “respective gene editing approaches” that were used in the 4 studies? Any differences in how well the approaches worked? Can the authors speculate as to the relative merits of these different approaches?

Since we have a full sub-section dedicated to genetic engineering for immune modulation, we have now referred the reader to section V in this sentence (line 199), where the specific mentioned/referenced editing approaches are described and discussed.

6. Line 193-5... what derivative cell types have been studied in the 4 references provided? Which of the 12 activating or inhibitory ligands were evaluated on which cell type? A little more color would be helpful here. eg, “For example, Smith et al assessed XX and found YY. Jones et al did ZZZ.”

We agree with the reviewer. We have now specified the ligands for each study and the respective findings in lines 204-210.

7. Line 198-210. This paragraph provides three specific examples and gives good comparative context. It shows the type of context that would be helpful in other sections. One small note, “PPRs written instead of PRRs”

We thank the reviewer for spotting this small error. The abbreviation has now been spelt out and corrected all throughout the text.

8. Line 259, what are the “discrepancies” between the two studies? Can you be more specific?

We have now specified the differences between the referred studies and re-wrote part of the sentence for clarity (lines 272-274).

9. Line 306-326. This section needs work. It is very dense. Can you break it up with a few specific examples for at least a couple of the assays being mentioned? What were the results? Which cell types have been studied and using which assays? What were key findings? Did any studies compare *in vivo* immunogenicity of two different cell types?

We agree with the reviewer that this particular paragraph was very dense. We have now displayed the *in vivo* experimental endpoints in a bullet form format to improve readability. Each point includes specific references that illustrate it but are not described due to word number limitation of the main text. Nonetheless, we have now added one exemplar study describing the studied cell type, the different assays used to study the immunogenicity of the cellular therapy *in vivo*, and the found key findings.

10. Line 485-6, what assays were used and what lineages were studied in the 4 referenced papers?

We have now added these informations in lines 516-517.

11. Line 488-498- what assays were used in some of the references cited and what were the results?

We have now added these informations for references 55 and 26 in lines 521-523 and 525-527, respectively.

12. Table 4, "Genetically Modified Cells" row, please ensure all the appropriate references that are mentioned in the text are also cited here (eg, missing ref 132 for immune cloaking?, please check)

We thank the reviewer for the comment. We have now checked all references and added the missing 132 related to "Genetically Modified Cells".